

# Universal Gibbons-Hawking-York term for theories
# with curvature, torsion and non-metricity

Johanna Erdmenger[1,2], Bastian Heß[1,2], Ioannis Matthaiakakis[1,2,3,4⋆] and René Meyer[1,2]

**1** Institute for Theoretical Physics and Astrophysics,
Julius-Maximilians-Universität Würzburg,
Am Hubland, D-97074 Würzburg, Germany
**2** Würzburg-Dresden Cluster of Excellence on Complexity
and Topology in Quantum Matter ct.qmat
**3** Dipartimento di Fisica, Università di Genova, via Dodecaneso 33, I-16146, Genova, Italy
**4** I.N.F.N. - Sezione di Genova, via Dodecaneso 33, I-16146, Genova, Italy

⋆ ioannis.matthaiakakis@edu.unige.it

## Abstract

Motivated by establishing holographic renormalization for gravitational theories with non-metricity and torsion, we present a new and efficient general method for calculating Gibbons-Hawking-York (GHY) terms. Our method consists of linearizing any nonlinearity in curvature, torsion or non-metricity by introducing suitable Lagrange multipliers. Moreover, we use a split formalism for differential forms, writing them in $(n-1)+1$ dimensions. The boundary terms of the action are manifest in this formalism by means of Stokes' theorem, such that the compensating GHY term for the Dirichlet problem may be read off directly. We observe that only those terms in the Lagrangian that contain curvature contribute to the GHY term. Terms polynomial solely in torsion and non-metricity do not require any GHY term compensation for the variational problem to be well-defined. We test our method by confirming existing results for Einstein-Hilbert and four-dimensional Chern-Simons modified gravity. Moreover, we obtain new results for torsionful Lovelock-Chern-Simons and metric-affine gravity. For all four examples, our new method and results contribute to a new approach towards a systematic hydrodynamic expansion for spin and hypermomentum currents within AdS/CFT.



# 1  Introduction

Curved Riemannian and pseudo-Riemannian spacetimes are of immediate relevance for astrophysics, cosmology and high energy physics. In most cases, the spacetime metric is the only dynamical field associated to the geometry, while further geometric structures are argued to be irrelevant by appealing to current experimental evidence. Recently, however, general spacetimes with non-vanishing curvature[1] $\Omega^\mu{}_\nu$ and torsion $T^\mu$ found applications in condensed matter systems—where lattice deformations generate non-trivial $\Omega^\mu{}_\nu$ and $T^\mu$ directly coupled to the electronic degrees of freedom—as a means to simulate high-energy phenomena in tabletop experiments [1–16]. In particular, torsion has been shown to alter the the hydrodynamic expansion by introducing terms relevant for both heavy-ion physics and electron flows in condensed matter [15–21].

We may, however, go even further beyond spacetimes with curvature and torsion to the more general metric-affine spacetimes hosting a non-trivial non-metricity one-form $Q_{\mu\nu}$. In condensed matter systems, this may be achieved by the introduction of dislocations and cuts in the lattice of a given material [22]. For our purposes, the importance of considering metric-affine spacetimes arises from the coupling of torsion and non-metricity to matter fields. Both torsion and non-metricity may be seen as the sources of conserved matter currents. In particular, $T^\mu$ is the source of the spin tensor $S_\mu{}^{\nu\rho}$, while $Q_{\mu\nu}$ is the source of the hypermomentum tensor $\Delta_{\rho\mu\nu}$ [23]. The spin tensor is well-known and has been studied in depth: In particular, spin transport was recently suggested to play an important role in hadron-hadron collisions at LHC [18,24,25], as well as in tabletop experiments on electronic systems [26,27]. In contrast,

---

[1]$\Omega^\mu{}_\nu$ is the curvature two-form, whose components are the Riemann tensor $R^\mu{}_{\nu\rho\sigma}$.

the hypermomentum tensor is not widely studied in the physics literature. The trace of this tensor, $J_\mu^D = \Delta_{\mu\nu}{}^\nu$, however, is referred to as dilation current and has found applications in the context of trace anomalies in condensed matter systems [3].

The source interpretation of torsion and non-metricity is the most relevant in the present context, due to its role in the AdS/CFT correspondence [28–30]. The AdS/CFT correspondence or holography is a duality between a weakly curved theory of gravity and a strongly coupled quantum field theory (QFT) living in one dimension lower, at the boundary of the curved spacetime. The sources coupled to the QFT determine the boundary conditions of the dynamical fields of the gravitational theory, while the on-shell gravitational action acts as the QFT's generating functional. Holography has widely been used to shed light on the transport properties of matter, for instance of the quark-gluon plasma or of graphene [31, 32]. These transport properties, however, concern only non-trivial energy-momentum and electric charge transfer in the QFT, i.e. only a dynamical metric and gauge field in the gravitational theory. It is our goal to extend the correspondence to include spin and hypermomentum transport in strongly coupled systems. The former arises from the intrinsic spin of particles, while the latter arises from the modification of the causal structure of spacetime due to matter [33]. Causality may exclude hypermomentum transport in relativistic systems of particle and astrophysics, since it may modify the light-cone structure and turn spacelike separated events into timelike separated ones under parallel transport. Condensed matter systems are not bound to such constraints, since their causal structure arises only as an emergent description of the system's electronic bands. We therefore expect hypermomentum to lead to new transport phenomena and perhaps novel phase transitions in this context.

In order to extend holography to the realm of both spin and hypermomentum transport, the dual gravitational theory must contain dynamical torsion and non-metricity tensors, i.e. we must consider dynamical theories of metric-affine gravity (MAG) on spacetimes with boundary.[2] The introduction of boundaries into the spacetime considered raises issues of fundamental importance that need to be addressed before applications can be considered. In particular, we have to ensure that the boundary conditions of the MAG fields are compatible with a well-defined variational problem. This is achieved in general relativity by the inclusion of the Gibbons-Hawking-York (GHY) boundary term.

The importance of the GHY boundary term in theories of gravity cannot be overstated. In the simplest of applications of gravitational theories, in terms of a Lagrangian, the GHY term makes the variational formulation well-posed [36–39]. In addition, within the Hamiltonian formalism, the GHY term allows us to correctly define the Hamiltonian as well as the asymptotic conserved charges of the theory, such as energy [38]. The GHY terms also play a crucial role within holography: First, the asymptotic charges evaluated at the boundary are precisely the charges of the dual QFT. Second, the on-shell gravitational action (and QFT generating functional) suffers from divergences. The GHY terms act as counterterms and provide (part of) the regularization necessary to define a finite gravitational action [40]. Thus, in order to use holography to derive QFT observables and understand spin and hypermomentum transport, we have to derive the GHY term for every MAG Lagrangian. We carry out this derivation in the present paper.

The GHY term we find provides the starting point for holographic renormalization [41] that we plan to address in the future in the present context. Holographic renormalization in turn will be the starting point for a fluid-gravity hydrodynamic expansion [42] including torsion and non-metricity. Also beyond holography, our results are of interest to both the general relativity and condensed matter communities. The first may use our results as a stepping stone towards analyzing extensions of general relativity to the MAG framework and their

---

[2]For early work on holographic models for spaces with torsion, but zero non-metricity, see [34, 35]. A recent discussion may be found in [16].

compatibility with cosmological data. For the second, in addition to forthcoming results on spin hydrodynamics that we expect to be obtainable based on the analysis presented here, our terms become important when considering the dynamics of defects that give rise to torsion and non-metricity on systems with boundaries. In this context, our GHY terms describe the dynamics of the defects on the boundary which are consistent with the dynamics in the bulk. Such terms may also become important when considering topological field theories on spaces with boundaries, perhaps with quantum anomalies, since they describe the emergent degrees of freedom at the boundary.

The main result of this work is a new and efficient method for deriving GHY terms for actions formulated in the language of differential forms. In particular, our generalized GHY term applies for any theory which is allowed to have curved, torsional and non-metric degrees of freedom in arbitrary polynomial combinations. We achieve this generality by formulating our result in terms of auxiliary fields which are calculated for generic theories by taking variations of their Lagrangian $n$-form with respect to suitable Lagrange multipliers. One of our main findings is that our generalized GHY term receives contributions from the variation of only those terms in the Lagrangian that contain curvature. Terms that are solely built from torsion and non-metricity do not contribute a GHY term, assuming that any derivatives of torsion and non-metricity have been converted into curvature polynomials by means of the Bianchi identities. Using our method we confirm the GHY term for the Einstein-Hilbert action in arbitrary dimensions. We furthermore verify a result for 4d Chern-Simons modified gravity up to a factor of 2. Moreover, we present new results for torsionful Lovelock-Chern-Simons and metric-affine gravity. These new results have the form which we expect from comparison of the Lagrangians with those of Einstein-Hilbert and 4d Chern-Simons modified gravity.

We begin the main part of this paper with section 2, where we briefly set up the geometric framework used. This allows us to present the main results of our work in a concise manner in the same section. Subsequently, we apply these general results to the special cases of Einstein-Hilbert and four-dimensional Chern-Simons modified gravity in section 3 to find familiar results as a check of our method. In the same section, we apply our method to the case of torsionful Lovelock-Chern-Simons gravity and derive new results for its GHY term. An explicit derivation of our results is given in section 4. The more involved case of MAG is considered in appendix A.

## 2 Geometric setup and summary of the main results

In the current section, we present our main result for the GHY term for any MAG theory in (12). To fully grasp the meaning of each term in (12), however, we must first give a lightning review of our geometric setup. For the interested reader, more details are given in section 4.

As mentioned in the introduction, the geometry that we consider in a dynamical setting is that of a metric-affine spacetime. This spacetime is an $n$-dimensional manifold $\mathcal{M}$ equipped with a coframe basis $\theta^{\mu}$, a metric $\mathrm{d}s^2 = g_{\mu\nu}\theta^{\mu} \otimes \theta^{\nu}$ and a connection one-form $\omega^{\mu}_{\nu} = \Gamma^{\mu}_{\rho\nu}\theta^{\rho}$, where Greek indices take values in $\{0, \ldots, n-1\}$. These fields are independent of each other and may be thought of as the kinematic variables of the spacetime in the same sense that a U(1) gauge field provides the kinematic variables for a U(1) gauge theory. Following this analogy further, we may define field strengths for $\theta^{\mu}$, $g_{\mu\nu}$ and $\omega^{\mu}_{\nu}$. To preserve diffeomorphism invariance, we construct these field strengths in terms of the exterior derivative $\mathrm{d}$ and the exterior covariant derivative $D = \mathrm{d} + \omega^{\mu}_{\nu}\rho(L)^{\nu}_{\mu}\wedge$, where $\rho(L)^{\nu}_{\mu}$ is the appropriate representation

of the GL(n,ℝ) generators $L$. In particular, the field strengths for the fields $\omega^\mu_\nu$, $\theta^\mu$, $g_{\mu\nu}$ are

the curvature two-form $\qquad \Omega^\mu{}_\nu := d\omega^\mu{}_\nu + \omega^\mu{}_\rho \wedge \omega^\rho{}_\nu = \frac{1}{2} R^\mu{}_{\nu\rho\sigma} \theta^\rho \wedge \theta^\sigma \,,$

the torsion two-form $\qquad T^\mu := D\theta^\mu = d\theta^\mu + \omega^\mu{}_\nu \wedge \theta^\nu = \frac{1}{2} T^\mu{}_{\rho\sigma} \theta^\rho \wedge \theta^\sigma \,,$ $\qquad$ (1)

the non-metricity one-form $\quad Q_{\mu\nu} := -Dg_{\mu\nu} = -(dg_{\mu\nu} - \omega^\rho{}_\mu g_{\rho\nu} - \omega^\rho{}_\nu g_{\mu\rho}) = Q_{\mu\nu\rho} \theta^\rho \,,$

respectively. The curvature $R^\mu{}_{\nu\rho\sigma}$ is the Riemann tensor, while $T^\mu{}_{\rho\sigma}$ is the torsion and $Q_{\mu\nu\rho}$ the non-metricity tensor. All three of the field strengths satisfy a corresponding Bianchi identity

$$
\begin{aligned}
D\Omega^\mu{}_\nu &= 0\,, \\
DT^\mu &= \Omega^\mu{}_\nu \wedge \theta^\nu\,, \\
DQ_{\mu\nu} &= \Omega_{\mu\nu} + \Omega_{\nu\mu}\,.
\end{aligned}
$$
$\qquad$ (2)

We introduce a boundary $\partial\mathcal{M}$ in $\mathcal{M}$ and a geometry on $\partial\mathcal{M}$ via two sets of vector fields. First, we define $\partial\mathcal{M}$ through the normal vector $n^\mu$, normalized as $n_\mu n^\mu = \varepsilon = -1$ for spacelike and $n_\mu n^\mu = \varepsilon = +1$ for timelike boundaries. Second, we introduce the vielbein $e^\mu_a$ as a basis of tangent vectors on $\partial\mathcal{M}$, $n_\mu e^\mu_a = 0$, where Latin indices take values in $\{0,\dots,n-2\}$. The vielbein and its dual $e^a_\mu$ induce a geometry on $\partial\mathcal{M}$ via pullback from $\mathcal{M}$. For example, the boundary coframe basis $\phi^a$ is obtained from the manifold coframe basis $\theta^\mu$ as $e^a_\mu \theta^\mu = \phi^a$, while the induced metric on $\partial\mathcal{M}$ is $\gamma_{ab} := e^\mu_a e^\nu_b g_{\mu\nu}$. While the metric and coframe project to the boundary via pullback, the connection is related to its boundary value $\omega^a_b$ in a more involved way. In particular, we assume that $\omega^\mu_\nu$ projects to $\omega^a_b$ via the vielbein postulate

$$
\omega^b_a = e^b_\mu \left( de^\mu_a + \omega^\mu{}_\nu e^\nu_a \right). \qquad (3)
$$

We prove the transformation property (3) within the MAG framework [23, 43] in section 4.3. To our knowledge this is the most general theory that includes curvature, torsion and non-metricity. It is thus natural to assume that (3) holds for a large range of theories featuring fewer fields or more symmetries.

Apart from the curvature, torsion and non-metricity associated to the projected connection, coframe and metric on the boundary, the embedding of $\partial\mathcal{M}$ in $\mathcal{M}$ allows for additional quantities that describe the geometry of $\partial\mathcal{M}$ relative to that of $\mathcal{M}$. For example, in the following we use extensively the extrinsic curvatures

$$
K^a := e^a_\mu Dn^\mu \qquad \text{and} \qquad \tilde{K}_a := e^\mu_a Dn_\mu = K_a - e^\mu_a n^\nu Q_{\mu\nu}. \qquad (4)
$$

The two definitions of the extrinsic curvature in (4) differ only by a deformation in the $n - e_a$ plane due to non-metricity. Thus, for metric compatible connections satisfying $Dg_{\mu\nu} = 0$ it suffices to consider only one of the extrinsic curvature definitions. The tensor components of $K^a$ tangent to the boundary give the extrinsic curvature familiar from the literature, $K^a{}_b = e^a_\mu e^\nu_b \nabla_\nu n^\mu$, where $\nabla_\nu$ is the covariant derivative on $\mathcal{M}$ [38, 44, 45].

This concludes our brief review of the geometry on $\mathcal{M}$ and the induced geometry on $\partial\mathcal{M}$. For investigating the dynamics of this geometry, we consider the most general action possible,

$$
S_{\text{orig}}[g_{\mu\nu}, \omega^\mu{}_\nu, \theta^\mu] = \int_\mathcal{M} \mathcal{L}(\Omega^\mu{}_\nu, T^\mu, Q_{\mu\nu})\,, \qquad (5)
$$

where the Lagrangian $n$-form $\mathcal{L}(\Omega^\mu{}_\nu, T^\mu, Q_{\mu\nu})$ is an arbitrary function of the curvature, torsion and non-metricity defined in (1). Not fixing a particular form for $\mathcal{L}$ ensures that our results hold for any action of interest and may be applied to a specific system by choosing $\mathcal{L}$ appropriately.

Note that we have restricted ourselves to actions which are polynomial in the field strengths but do not involve their derivatives. This is not a severe restriction since the Bianchi identities (2) tell us that derivatives of curvature vanish, while those of torsion and non-metricity can be reduced to polynomial terms in curvature.

Our goal is to make the variation of the action (5) well-defined by adding the appropriate boundary terms to it. To find these terms we vary $S_{\text{orig}}$ and isolate the terms on the boundary that do not vanish after enforcing the boundary conditions on the fields. To be precise, the boundary conditions we consider are $\delta g_{\mu\nu}\big|_{\partial\mathcal{M}} = 0$, $\delta\theta^{\mu}\big|_{\partial\mathcal{M}} = 0$ and $\delta\omega^{\mu}_{\nu}\big|_{\partial\mathcal{M}} = 0$. However, according to [38] it suffices to demand that the Dirichlet boundary conditions

$$\delta\gamma_{ab} = 0, \quad \delta\phi^{a} = 0, \quad \delta\omega^{a}_{b} = 0, \tag{6}$$

hold since the original conditions may be reinstated by gauge transformations on $\partial\mathcal{M}$.

In principle, we may now obtain the boundary terms from a direct variation of the action $S_{\text{orig}}$. However, general covariance requires that we write the boundary terms in terms of geometric quantities on $\partial\mathcal{M}$. To achieve this we need to perform a $3+1$ decomposition[3] of $S_{\text{orig}}$ which is impossible to write down if $\mathcal{L}$ is not specified. We circumvent this issue by the method of Lagrange multipliers expounded in [46] for the case of $\mathcal{L}(\Omega^{\mu}_{\nu})$ gravity. This method essentially makes the action linear in the field strengths without losing any information of the dynamics induced by $\mathcal{L}$. In order to isolate the boundary terms, we introduce the Lagrange multipliers $\varphi_{\mu}{}^{\nu}$, $t_{\mu}$ and $q^{\mu\nu}$ and turn the Langrangian into a linear function of $\Omega^{\mu}_{\nu}$, $T^{\mu}$ and $Q_{\mu\nu}$. In particular, we consider the gravitational action

$$S[g_{\mu\nu}, \omega^{\mu}_{\nu}, \theta^{\mu}, \varphi^{\mu}_{\nu}, \varrho^{\mu}_{\nu}, t_{\mu}, \tau^{\mu}, q^{\mu\nu}, \sigma_{\mu\nu}] = \int_{\mathcal{M}} \Big[ \mathcal{L}(\varrho^{\mu}_{\nu}, \tau^{\mu}, \sigma_{\mu\nu}) + *\varphi_{\mu}{}^{\nu} \wedge (\Omega^{\mu}_{\nu} - \varrho^{\mu}_{\nu}) \tag{7}$$
$$+ *t_{\mu} \wedge (T^{\mu} - \tau^{\mu}) + *q^{\mu\nu} \wedge (Q_{\mu\nu} - \sigma_{\mu\nu}) \Big],$$

where $\varrho^{\mu}_{\nu}$, $\tau^{\mu}$ and $\sigma_{\mu\nu}$ are auxiliary fields and $*$ is the Hodge duality. Expressing $S$ as in (7) allows us to directly access $\Omega^{\mu}_{\nu}$, $T^{\mu}$ and $Q_{\mu\nu}$ regardless of the explicit form of $\mathcal{L}$. We choose the Lagrange multipliers to be independent of each other and of the fields $\omega^{\mu}_{\nu}$, $\theta^{\mu}$ and $g_{\mu\nu}$. In this way we ensure that the equations of motion for $\Omega^{\mu}_{\nu}$, $T^{\mu}$ and $Q_{\mu\nu}$ are kept unchanged if we first impose the equations of motion

$$\Omega^{\mu}_{\nu} = \varrho^{\mu}_{\nu}, \qquad T^{\mu} = \tau^{\mu}, \qquad Q_{\mu\nu} = \sigma_{\mu\nu}, \tag{8}$$

for the Lagrange multipliers $\varphi_{\mu}{}^{\nu}$, $t_{\mu}$ and $q^{\mu\nu}$, respectively. To that end, we additionally demand these Lagrange multipliers to have the exact same symmetries as the corresponding field strengths.

In order to express the boundary terms in terms of a diffeomorphic invariant Lagrangian on the boundary, we consider projections of the Lagrange multipliers on $\partial\mathcal{M}$. In particular, we project each index of the Lagrange multipliers either to the boundary by contraction with the vielbein $e^{a}_{\mu}$ or normal to the boundary by contraction with $n^{\mu}$. We abbreviate these contractions as $\varphi_{a\mathbf{n}} := e^{\mu}_{a} n^{\nu} \varphi_{\mu\nu}$ for instance. These projections may be regarded as an extension of the $3+1$ decomposition in general relativity and can be derived by expanding every contraction of Greek indices in (7) using

$$\delta^{\mu}_{\nu} = e^{\mu}_{a} e^{a}_{\nu} + \varepsilon n^{\mu} n_{\nu}. \tag{9}$$

For example $A^{\mu} \wedge B_{\mu} = A^{a} \wedge B_{a} + \varepsilon A_{\mathbf{n}} \wedge B_{\mathbf{n}}$ for generic differential forms $A^{\mu}$, $B_{\mu}$. The explicit form of the action $S$ in terms of the $3+1$ decomposition is rather lengthy and not that enlightening.

---

[3]While we work in general $n$ dimensions and perform a full $(n-1)+1$ decomposition, we use the term $3+1$ decomposition for compactness throughout.

So we leave (7) unchanged and keep in mind that for further calculations any of the Greek indices is $3+1$ decomposed as just described.

While the equations of motion (8) of the Lagrange multipliers $\varphi_\mu{}^\nu$, $t_\mu$ and $q^{\mu\nu}$ ensure the equivalence of $S$ and $S_{\text{orig}}$, the equations of motion of the additional fields $\varrho^\mu{}_\nu$, $\tau^\mu$ and $\sigma_{\mu\nu}$ yield constraints which enable us to determine the explicit forms of $\varphi_\mu{}^\nu$, $t_\mu$ and $q^{\mu\nu}$ in terms of $\mathcal{L}$. Among these constraints the ones which yield $*\varphi^{\mathbf{n}a} := n^\alpha e^a_\beta * \varphi_\alpha{}^\beta$, $*\varphi_{a\mathbf{n}} := e^\alpha_a n_\beta * \varphi_\alpha{}^\beta$ and $*\varphi_{\mathbf{nn}} := n^\alpha n_\beta * \varphi_\alpha{}^\beta$ are particularly important for us. These are

$$
\begin{aligned}
*\varphi^{\mathbf{n}a} \wedge \delta\varrho_{\mathbf{n}a} &= \varepsilon\delta_{\varrho_{\mathbf{n}a}}\mathcal{L}(\varrho_{\mathbf{n}a}, \varrho^{a\mathbf{n}}, \varrho_{\mathbf{nn}}, \dots), \\
*\varphi_{a\mathbf{n}} \wedge \delta\varrho^{a\mathbf{n}} &= \varepsilon\delta_{\varrho^{a\mathbf{n}}}\mathcal{L}(\varrho_{\mathbf{n}a}, \varrho^{a\mathbf{n}}, \varrho_{\mathbf{nn}}, \dots), \\
*\varphi^{\mathbf{nn}} \wedge \delta\varrho_{\mathbf{nn}} &= \delta_{\varrho_{\mathbf{nn}}}\mathcal{L}(\varrho_{\mathbf{n}a}, \varrho^{a\mathbf{n}}, \varrho_{\mathbf{nn}}, \dots),
\end{aligned}
\tag{10}
$$

obtained from varying (7) with respect to $\varrho_{\mathbf{n}a}$, $\varrho_{a\mathbf{n}}$ and $\varrho_{\mathbf{nn}}$.[4] To put it simple, $\varphi^\mu{}_\nu$ is the Hodge dual of the equations of motion for $\Omega^\mu{}_\nu$.[5]

In addition to the $3+1$ decomposition of the Lagrange multipliers and auxiliary fields we also need to introduce the $3+1$ decomposition of the curvature, torsion and non-metricity into the action $S$ in (7). The full details of the derivation are given in section 4. Here we mention the $3+1$ decomposition of curvature which plays an important role in the examples we consider in section 3. We have

$$
\begin{aligned}
e^a_\mu e^\nu_b \Omega^\mu{}_\nu &= \Omega^a{}_b - \varepsilon K^a \wedge \tilde{K}_b, & n_\mu e^\nu_a \Omega^\mu{}_\nu &= D\tilde{K}_a + \frac{\varepsilon}{2}\tilde{K}_a \wedge Q_{\mathbf{nn}}, \\
e^a_\mu n^\nu \Omega^\mu{}_\nu &= DK^a + \frac{\varepsilon}{2}K^a \wedge Q_{\mathbf{nn}}, & n_\mu n^\nu \Omega^\mu{}_\nu &= \frac{1}{2}DQ_{\mathbf{nn}} + K^a \wedge \tilde{K}_a,
\end{aligned}
\tag{11}
$$

where we define $\Omega^a{}_b := \mathrm{d}\omega^a_b + \omega^a_c \wedge \omega^c_b$ for the curvature on the boundary.[6] The result (11) may be considered a generalized version of the Gauß-Codazzi equations.

Finally, let us mention the strategy we use to obtain the GHY term for the generic action (5) after the $3+1$ decomposition has been made. Expressing the action in a $3+1$ form results in two types of terms appearing in the Lagrangian; one containing the fields $\omega^\mu_\nu$, $\theta^\mu$, $g_{\mu\nu}$ and their projection to the boundary and another one containing their derivatives. Due to the Dirichlet boundary conditions (6) the first type does not contribute at the boundary and, thus, only the second type of terms can lead to boundary contributions under variation. To calculate them, we employ Stokes' theorem to rewrite these terms as a pure boundary integral. We then vary the boundary integral and isolate the terms that do not vanish after variation. These are the terms which have to be subtracted from $S$ by means of the GHY term. Carrying out this calculation in section 4 we find the explicit form of the GHY term for a MAG theory with Lagrangian $\mathcal{L}$ to be

$$
\begin{aligned}
S_{\text{GHY}} &= -\int_{\partial\mathcal{M}} \left( -\varepsilon\tilde{K}_a \wedge *\varphi^{\mathbf{n}a} + \varepsilon K^a \wedge *\varphi_{a\mathbf{n}} + \frac{1}{2}Q_{\mathbf{nn}} \wedge *\varphi_{\mathbf{nn}} \right)\Big|_{\partial\mathcal{M}} \\
&= -\int_{\partial\mathcal{M}} \left( \varepsilon K^a \wedge *(\varphi_{a\mathbf{n}} - \varphi_{\mathbf{n}a}) + \varepsilon Q_{\mathbf{n}a} \wedge *\varphi^{\mathbf{n}a} + \frac{1}{2}Q_{\mathbf{nn}} \wedge *\varphi_{\mathbf{nn}} \right)\Big|_{\partial\mathcal{M}},
\end{aligned}
\tag{12}
$$

where $\varphi_{a\mathbf{n}}$, $\varphi_{\mathbf{n}a}$ and $\varphi_{\mathbf{nn}}$ are implicitly expressed in terms of $\mathcal{L}$ through the constraints (10).

---

[4]Only these variations are relevant, since the boundary conditions (6) do not fix $\omega^a_\mathbf{n}$, $\omega^\mathbf{n}_a$ and $\omega^\mathbf{n}_\mathbf{n}$.

[5]Note that these are not Einstein's equations which result from a variation with respect to the connection.

[6]Note the components of $\Omega^a{}_b$ also include normal contributions in general. The purely hypersurface curvature, $\Omega^{(n-1)a}{}_b$, is found by means of projecting the components of $\Omega^a{}_b$ to the hypersurface, that is $\Omega^{(n-1)a}{}_b = \Omega^a{}_b(\varphi_c, \varphi_d)\phi^c \wedge \phi^d$, with $\varphi_a$ the dual of $\phi^a$ (see (44)).

A couple of comments about $S_{\text{GHY}}$ are in order. First, we note that our result should be expected. All terms in (12) depend on the first derivatives of the metric as in typical general relativity. That torsion does not appear explicitly in the GHY terms should also be expected. Torsion may contribute boundary terms only via the derivative of the frame field it contains, see (1). This derivative measures the non-holonomicity of the frame and is not a true geometric invariant of the theory. Therefore, in a generally covariant theory, vanishing non-holonomicity is locally enforceable and as a result the torsion two-form is only a polynomial in $\omega^{\mu}_{\nu}$ and $\theta^{\mu}$. This is why the torsion two-form cannot appear explicitly in the GHY terms. Of course, this does not forbid turning the GHY term from a function of the curvature to a function of torsion if additional constraints between the field strengths are taken into account as in teleparallel theories of gravity for example [47, 48].

Second, in the important case of vanishing non-metricity, $Q_{\mu\nu} = 0$, the GHY term (12) simplifies considerably to

$$S_{\text{GHY}}^{Q=0} = 2 \int_{\partial\mathcal{M}} \varepsilon K^a \wedge *\varphi_{\mathbf{n}a}|_{\partial\mathcal{M}} \,. \tag{13}$$

That is, the GHY term is a direct generalization of the GHY term of general relativity which is proportional to the extrinsic curvature of the boundary.

Finally, some technical notes are in place. First, for the calculation we considered $\varrho_{\mathbf{n}a}$ and $\varrho_{a\mathbf{n}}$ as independent although curvature fulfills $\Omega_{\mu\nu} + \Omega_{\nu\mu} = DQ_{\mu\nu}$ according to (2) and $\varrho_{\mu\nu}$ has the same symmetry as $\Omega_{\mu\nu}$. This simplifies our calculation considerably without altering the final result. Alternatively, it is possible to invoke the symmetry relation satisfied by $\varrho_{\mu\nu}$ prior to the variation. In appendix B we explicitly show that this yields the same resulting GHY term (13). Second, we worked entirely in the first order formalism where all fields are independent. If one wants to consider a second order one, the equations of motion must be used to express all fields in terms of the independent ones. In traditional general relativity, this involves solving for $\omega^{\mu}_{\nu}$ in terms $g_{\mu\nu}$. Substituting the solution into $\mathcal{L}$ yields a new Lagrangian $\tilde{\mathcal{L}}$ for the independent fields. If $\tilde{\mathcal{L}}$ is a function of the remaining independent field strengths, say $\tilde{\mathcal{L}} = \tilde{\mathcal{L}}(\Omega^{\mu}_{\ \nu})$, our algorithm may still be applied mutatis mutandis to obtain the GHY term for $\tilde{\mathcal{L}}$. We have implicitly used this observation when writing down (13) for the GHY terms of metric compatible theories in the following section.

To recap, the calculation of the GHY term for *any* MAG theory follows these steps: First, choose the Lagrangian $\mathcal{L}$ of the MAG of interest. Second, express $\mathcal{L}$ in a $3+1$ decomposed form (see section 4). Third, derive the Lagrange multiplier $\varphi_{\mu\nu}$ through the constraint equations (10). Finally, substitute the result into the general form (12) of the GHY term. In the following section, we apply this algorithm to several particular theories of gravity as a consistency check of our results.

## 3 Examples for Gibbons-Hawking-York terms

In the current section we apply the algorithm explained in the previous section to calculate the GHY terms for several theories of gravity. The examples we consider involve the Einstein-Hilbert action in general dimensions, the action of 4d Chern-Simons modified gravity and the 5d Lovelock-Chern-Simons action. We also calculate the GHY term for the most general MAG Lagrangian quadratic in the field strengths. Since both the calculation and the result for the MAG GHY terms are quite lengthy, but do not add any additional insight regarding the formalism, we relegate them to appendix A.

## 3.1 Einstein-Hilbert gravity $\mathcal{L} \propto R$

As a very basic consistency check of our algorithm, we first consider the Einstein-Hilbert action

$$S^{\mathrm{EH}} = \frac{1}{2\kappa} \int \mathrm{d}^4 x \sqrt{-g} R = \frac{1}{2\kappa} \int \eta^{\mu\nu} \wedge \Omega_{\mu\nu}, \tag{14}$$

where $\eta^{\mu\nu} := *(\theta^\mu \wedge \theta^\nu)$. The Lagrangian for this action is $\mathcal{L} = \mathcal{L}(\Omega^\mu{}_\nu) = \eta^{\mu\nu} \wedge \Omega_{\mu\nu}/2\kappa$. Hence, the general form of the extended action (7) including the auxiliary fields and Lagrange multipliers reads

$$S^{\mathrm{EH}}_{\mathrm{aux}} = \int \left( \frac{1}{2\kappa} \eta^{\mu\nu} \wedge \varrho_{\mu\nu} + *\varphi_\mu{}^\nu \wedge (\Omega^\mu{}_\nu - \varrho^\mu{}_\nu) \right), \tag{15}$$

in the case of Einstein-Hilbert gravity. The relevant terms in the $3+1$ decomposition of $\mathcal{L}(\varrho_{\mu\nu})$ for our calculation of the GHY term are

$$\mathcal{L}(\varrho_{\mu\nu}) = \frac{1}{2\kappa} \eta^{\mu\nu} \wedge \varrho_{\mu\nu} \simeq \frac{1}{2\kappa} \varepsilon \eta_{\mathbf{n}a} \wedge \varrho^{\mathbf{n}a}, \tag{16}$$

where $\simeq$ omits irrelevant terms of the decomposition. According to (10), the equations of motion for $\varrho^{\mathbf{n}a}$ read

$$*\varphi_{\mathbf{n}a} \wedge \delta \varrho^{\mathbf{n}a} = \varepsilon \delta_{\varrho^{\mathbf{n}a}} \mathcal{L}(\varrho) = \frac{1}{2\kappa} \eta_{\mathbf{n}a} \wedge \delta \varrho^{\mathbf{n}a}, \tag{17}$$

which fix the form of $\varphi_{\mathbf{n}a}$ to

$$*\varphi_{\mathbf{n}a} = \frac{1}{2\kappa} \eta_{\mathbf{n}a}. \tag{18}$$

Substituting (18) into our general result (13) for the GHY term, we obtain

$$S^{\mathrm{EH}}_{\mathrm{GHY}} = 2 \int_{\partial \mathcal{M}} \varepsilon K^a \wedge *\varphi_{\mathbf{n}a}|_{\partial \mathcal{M}} = \frac{\varepsilon}{\kappa} \int_{\partial \mathcal{M}} \mathrm{d}^3 x \sqrt{|\gamma|} K^a{}_a, \tag{19}$$

which is the well-known result for the GHY term in general relativity [36, 37]. Note that we used the 4d Einstein-Hilbert action, but since the differential geometric formulation of the action (14) holds for an arbitrary number of dimensions, our result is actually more general. In particular, the GHY term

$$S^{\mathrm{EH}}_{\mathrm{GHY}} = \frac{\varepsilon}{\kappa} \int_{\partial \mathcal{M}} \mathrm{dVol}_{\partial \mathcal{M}} \sqrt{|\gamma|} K^a{}_a, \tag{20}$$

solves the Dirichlet problem for Einstein-Hilbert gravity on any $n$-manifold $\mathcal{M}$. The abbreviation $\mathrm{dVol}_{\partial \mathcal{M}} := \phi^0 \wedge \cdots \wedge \phi^{n-2}$ is equal to the standard integral measure $\mathrm{d}^{n-1} y$ in holonomic coordinates $\phi^a = \mathrm{d} y^a$.

## 3.2 4d Chern-Simons modified gravity $\mathcal{L} \propto \Omega^2$

As a next check of our formalism we turn to the four-dimensional Chern-Simons action as used in Chern-Simons modified gravity [49]. This provides a non-trivial check since the action is quadratic in curvature and the GHY term was already derived in [50].

The Chern-Simons part of the full action is[7]

$$S^{\text{CS}} = \frac{\kappa}{4} \int d^4 x \sqrt{-g}\, \theta * RR = \frac{\kappa}{2}(-1)^{\text{indg}} \int \theta\, \Omega^{\mu}{}_{\nu} \wedge \Omega^{\nu}{}_{\mu}\,, \tag{21}$$

where the background scalar field $\theta$ must not be confused with the coframe basis $\theta^{\mu}$. We read off $\mathcal{L}(\Omega^{\mu}{}_{\nu}) = \frac{\kappa}{2}(-1)^{\text{indg}}\theta\, \Omega^{\mu}{}_{\nu} \wedge \Omega^{\nu}{}_{\mu}$ from the action (21) so that the equation of motion (10) for $\varrho^{\mathbf{n}a}$ yields

$$* \varphi_{\mathbf{n}a} \wedge \delta \varrho^{\mathbf{n}a} = \varepsilon \delta_{\varrho^{\mathbf{n}a}} \mathcal{L} = (-1)^{\text{indg}} \kappa \theta \varrho_{a\mathbf{n}} \wedge \delta \varrho^{\mathbf{n}a}\,, \tag{22}$$

which leads to $* \varphi^{\mathbf{n}a} = (-1)^{\text{indg}} \kappa \theta \varrho^{a\mathbf{n}}$.

We observe that $\varphi^{\mathbf{n}a}$ depends explicitly on $\varrho^{a\mathbf{n}}$, unlike the Einstein-Hilbert action. To proceed, we use the equations of motion (8) of $\varphi^{\mu\nu}$ to fix $\varrho_{\mu\nu} = \Omega_{\mu\nu}$. Subsequently, we employ the $3+1$ decomposition of curvature which is given by the Gauß-Codazzi equations (11) to express $\varphi^{\mathbf{n}a}$ in terms of geometric quantities on $\partial\mathcal{M}$. We find

$$* \varphi^{\mathbf{n}a}\big|_{\varrho_{\mu\nu}=\Omega_{\mu\nu}} = (-1)^{\text{indg}} \kappa \theta \varrho^{a\mathbf{n}}\big|_{\varrho_{\mu\nu}=\Omega_{\mu\nu}} = (-1)^{\text{indg}} \kappa \theta D K^a\,, \tag{23}$$

which evaluated on the boundary reads

$$* \varphi^{\mathbf{n}a}\big|_{\partial\mathcal{M}} = (-1)^{\text{indg}} \kappa \theta \nabla_a K^c{}_b\, \phi^a \wedge \phi^b\,. \tag{24}$$

Here we define the boundary covariant derivative as $\nabla_a K^c{}_b := \partial_a K^c{}_b + \Gamma^c_{ad} K^d{}_b - \Gamma^d_{ab} K^c{}_d$, where the connection coefficients are related to the boundary connection (3) as $\omega^a_b = \Gamma^a_{cb} \phi^c$. To match the conventions of [50] we choose $|g| = -g$, $|\gamma| = \gamma$ as well as torsion freedom. This, in conjunction with (13), leads to the GHY term

$$S^{\text{CS}}_{\text{GHY}} = 2\kappa \int_{\mathcal{M}_3} d^3 x \sqrt{\gamma}\, \theta \epsilon^{ijk} K_i{}^l \nabla_j K_{kl}\,, \tag{25}$$

where $\epsilon^{ijk}$ denotes the totally antisymmetric tensor which differs from the $\varepsilon^{ijk}$-symbol by a factor of $1/\sqrt{\gamma}$. The result (25) coincides with the result of [50] up to a factor of 2. However, a calculation of this GHY term via the methods of [46] and [39] supports our result (25) including the factor of 2. We now turn to the derivation of the GHY term for torsionful Lovelock-Chern-Simons gravity which is not known in literature as far as we are aware.

## 3.3 Lovelock-Chern-Simons gravity

For our final example, we consider Lovelock-Chern-Simons gravity in a 5d spacetime with boundary. This example involves both a non-trivial polynomial Lagrangian for curvature and non-vanishing torsion. In particular, we investigate the action

$$S^{\text{LCS}} = \kappa \int_{\mathcal{M}_5} \epsilon_{\mu\nu\alpha\beta\sigma} \left( \Omega^{\mu\nu} \wedge \Omega^{\alpha\beta} + \frac{2}{3} \Omega^{\mu\nu} \wedge \theta^{\alpha} \wedge \theta^{\beta} + \frac{1}{5} \theta^{\mu} \wedge \theta^{\nu} \wedge \theta^{\alpha} \wedge \theta^{\beta} \right) \wedge \theta^{\sigma}\,, \tag{26}$$

employed in [16, 51]. As in the previous examples we identify the integrand in (26) as $\mathcal{L}$ and use the equations of motion (8), (10) for $\varphi^{\mu}{}_{\nu}$ and $\varrho^{\mu}{}_{\nu}$ to obtain

$$* \varphi_{\mathbf{n}a}\big|_{\varrho_{\mu\nu}=\Omega_{\mu\nu}} = 2\varepsilon\kappa\epsilon_{\mu\nu\alpha\beta\sigma} n^{\mu} e_a^{\nu} \left( \Omega^{\alpha\beta} + \frac{1}{3}\theta^{\alpha} \wedge \theta^{\beta} \right) \wedge \theta^{\sigma}\,. \tag{27}$$

---

[7]In the differential geometric version of (21) we could alternatively use a notation which involves the Hodge dual of one of the curvature two-forms. We refrain from that since the variation of Hodge duals is non-trivial. For details on the variation of Hodge duals as well as interior products and their inclusion into the framework of this paper, see appendix A.1.

Note that since $\epsilon_{\mu\nu\alpha\beta\sigma}$ in (27) is contracted with $n^\mu$, the $\alpha, \beta$ and $\sigma$ indices are all on $\partial\mathcal{M}$ which means that they need to be contracted with $e_a^\mu$ when evaluating the $3+1$ decomposition of $\varphi_{\mathbf{n}a}$. Thus, we may write (27) as

$$* \varphi_{\mathbf{n}a}\big|_{\varrho_{\mu\nu}=\Omega_{\mu\nu}} = 2\varepsilon\kappa\epsilon_{abcd}\left(e_\alpha^b e_\beta^c \Omega^{\alpha\beta} + \frac{1}{3}\phi^b \wedge \phi^c\right) \wedge \phi^d. \tag{28}$$

Employing the Gauß-Codazzi equations (11) for vanishing non-metricity, we evaluate $\varphi_{\mathbf{n}a}$ in terms of geometric quantities on the boundary as

$$* \varphi_{\mathbf{n}a}\big|_{\mathcal{M}_4} = 2\varepsilon\kappa\epsilon_{abcd}\left(\Omega^{bc} - \varepsilon K^b \wedge K^c + \frac{1}{3}\phi^b \wedge \phi^c\right) \wedge \phi^d\bigg|_{\mathcal{M}_4}, \tag{29}$$

where $\Omega^a{}_b := d\omega_b^a + \omega_c^a \wedge \omega_b^c$ was defined in (11). By virtue of (13) the GHY term of torsionful Lovelock-Chern-Simons gravity is then

$$S_{\mathrm{GHY}}^{\mathrm{LCS}} = -4\kappa\int_{\mathcal{M}_4}\epsilon_{abcd}\left(\Omega^{ab} - \varepsilon K^a \wedge K^b + \frac{1}{3}\phi^a \wedge \phi^b\right) \wedge \phi^c \wedge K^d\bigg|_{\mathcal{M}_4}. \tag{30}$$

In components (30) takes the form

$$S_{\mathrm{GHY}}^{\mathrm{LCS}} = -4\kappa\int_{\mathcal{M}_4}\mathrm{dVol}_{\mathcal{M}_4}\sqrt{-\gamma}\left[3!\left(\frac{1}{2}R^{ab}{}_{[ab} - \varepsilon K^a{}_{[a}K^b{}_b\right)K^c{}_{c]} + 2K^a{}_a\right]. \tag{31}$$

General considerations regarding (30) show it is consistent with expectations. To see this, consider the Lovelock-Chern-Simons action (26) in more detail. This action consists of three terms. The first of them which is a quadratic curvature term in five dimensions is new to us so far. Because of its quadratic curvature form we expect it to yield a GHY term of a new form which is of higher order in the extrinsic curvature just as we observed it in the example of 4d Chern-Simons modified gravity in section 3.2. We immediately observe this behaviour from the generalized Gauß-Codazzi equations (11) which give the curvature $3+1$ decomposition. The third term in (26) has no curvature contribution and is not expected to yield a GHY contribution since these contributions all relate to variations of curvature, see (12). The remaining second term we already know. In fact, we rewrite it as

$$S^{\mathrm{LCS, 2nd\ term}} = \kappa\int_{\mathcal{M}_5}\epsilon_{\mu\nu\alpha\beta\sigma}\frac{2}{3}\Omega^{\mu\nu} \wedge \theta^\alpha \wedge \theta^\beta \wedge \theta^\sigma = 8\kappa\int_{\mathcal{M}_5}\eta^{\mu\nu} \wedge \Omega_{\mu\nu}, \tag{32}$$

where $\eta^{\mu\nu} := *(\theta^\mu \wedge \theta^\nu)$ was defined in section 3.1. Note that this is nothing but a 5-dimensional Einstein-Hilbert action. In section 3.1 we already noticed that actions of this form always have the same GHY term regardless of their dimension. Hence, we expect a contribution to the total Lovelock-Chern-Simons GHY term of the type

$$S_{\mathrm{GHY}}^{\mathrm{LCS, 2nd\ term}} \propto \int_{\partial\mathcal{M}}\mathrm{dVol}_{\partial\mathcal{M}}\sqrt{|\gamma|}K^a{}_a. \tag{33}$$

We only write a proportionality instead of an equality in the latter equation since there could be additional contributions from the quadratic curvature term which we omit in this consistency consideration. We observe all of these expected contributions in our result (31).

For vanishing torsion, the GHY term for Lovelock-Chern-Simons theory was derived in [52] following the method of dimensional continuation of [53]. Our results agree with those in [52] up to a factor of 3 in the cubic extrinsic curvature term in (30). Unfortunately, such a discrepancy should be expected since—as the authors of [52] stress—the dimensional continuation

method cannot be used if torsion is non-vanishing. The reason for this is quite simple: The authors of [52] and [53] essentially use the two first rungs of the descent ladder for gravitational anomalies in the absence of torsion [54], in order to define their respective GHY terms. The inclusion of torsion modifies the descent procedure—and hence the GHY terms—by modifying the BRS algebra of the fields [55]. It would be interesting to use the results of the descent procedure described in [55] to explicitly check that our result (30) for the Lovelock-Chern-Simons GHY term can be derived analogously to [52] in the case of vanishing torsion. In this sense, (30) may be regarded as a generalization of the Lovelock-Chern-Simons GHY term in [52] to torsionful Lovelock-Chern-Simons theory. Moreover, our formalism does not need to introduce a background manifold for deriving the GHY term, which may be considered another generalization of [52,53]. Some results regarding the GHY term for torsionful Lovelock-Chern-Simons theory also exist in the literature. Namely, the action of Lovelock-Chern-Simons (26) was renormalized in [51]. To achieve this, the authors of [51] considered a finite Fefferman-Graham expansion and compensated the divergent terms by addition of counterterms to the action. The non-divergent terms in this procedure were not given explicitly in [51] but calculated in [16], where they were interpreted as the GHY term of Lovelock-Chern-Simons gravity. This term constitutes an on-shell result whereas our result (30) is the universal GHY term for the Lovelock-Chern-Simons action (26). From this point of view, our result (30) may be considered as a generalization of the Lovelock-Chern-Simons GHY term in [16].

Our results for the GHY term of torsionful Lovelock-Chern-Simons theory constitute a new result in the literature. An additional novel result can be found in appendix A, where we derive the GHY term for the *most general* MAG Lagrangian quadratic in the fields. In the same appendix we include details on how to treat interior products and Hodge duals under the variations necessary to apply our algorithm. These allow to work directly in the differential form version of the Lagrangian, which usually provides the most economical expression for the Lagrangian. Before proceeding to appendix A, we suggest reading section 4. This section contains the explicit derivation of the general formula for the GHY term (12) along with some useful formulae regarding the 3 + 1 decomposition of the field strengths.

# 4 Derivation of the Gibbons-Hawking-York term

In this section we present the explicit calculation leading to the GHY term (12) given in section 2. The essence of the calculation of the GHY term involves the 3 + 1 decomposition of the fundamental fields, $\omega^{\mu}_{\ \nu}$, $g_{\mu\nu}$ and $\theta^{\mu}$, and their field strengths, $\Omega^{\mu}_{\ \nu}$, $Q_{\mu\nu}$ and $T^{\mu}$ respectively. In order to derive such a decomposition, we first consider the geometric setup for spacetime foliations introduced in section 2 in more detail. This involves an investigation of how the frame fields $\theta^{\mu}$, $n^{\mu}$ and $e^{a}_{\mu}$ decompose with respect to the spacetime foliation.

## 4.1 Frame decomposition

In this subsection, we first introduce frame decompositions on manifolds in the vielbein formalism. These frame decompositions constitute the basis for the subsequent investigation of foliations in the 3 + 1 formalism [44, 45, 56], in which the manifold is described as a stack of hypersurfaces. We assume that neither torsion nor non-metricity vanishes and use differential forms throughout to simplify the analysis (see e.g. [23, 43, 57]).

We consider an $n$-dimensional manifold $\mathcal{M}$ equipped with a generic coframe $\theta^{\mu}$, $\mu \in \{0, \dots, n-1\}$. Locally, a coframe is a basis of the cotangent space at each point $p$ of $\mathcal{M}$. In order to describe how the boundary is embedded in $\mathcal{M}$, we aim at decomposing all geometric quantities on $\mathcal{M}$ into contributions tangent and normal to the boundary. For this reason,

we assume that $\mathcal{M}$ is foliated by a family of hypersurfaces $\{\Sigma_\lambda\}_\lambda$, where $\lambda =$const defines each hypersurface $\Sigma_\lambda$. In order to be able to describe the embedding of the boundary as well as the boundary components of the manifold's geometric quantities torsion, curvature, and non-metricity, we choose this foliation such that the asymptotic boundary $\partial\mathcal{M}$ is one of the hypersurfaces. The most basic quantity to decompose in this foliation is the coframe $\theta^\mu$. We associate a frame field $\vartheta_\mu$ to this coframe by means of $\theta^\mu(\vartheta_\nu) = \delta_\nu^\mu = \vartheta_\nu(\theta^\mu)$, and decompose both according to

$$\theta^\mu = e_a^\mu \tilde{\phi}^a + t^\mu \phi \,, \tag{34a}$$

$$\vartheta_\mu = E_\mu^a \varphi_a + \frac{\varepsilon}{N} n_\mu \tilde{\varphi} \,, \tag{34b}$$

where $a \in \{0, \ldots, n-2\}$. The new tensors introduced in this decomposition are given by

$$e_a^\mu := \theta^\mu(\varphi_a), \qquad E_\mu^a := \vartheta_\mu(\tilde{\phi}^a), \qquad t^\mu := \theta^\mu(\tilde{\varphi}). \tag{35}$$

Using the *induced metric* on $\Sigma_\lambda$, $\gamma_{ab} := e_a^\mu e_b^\nu g_{\mu\nu}$, we define

$$\varepsilon := \mathrm{sgn}\left(\frac{\det g_{\mu\nu}}{\det \gamma_{ab}}\right), \qquad N := 1/\sqrt{|g^{\mu\nu}\vartheta_\mu(\phi)\vartheta_\nu(\phi)|}, \qquad n_\mu := \varepsilon N \vartheta_\mu(\phi). \tag{36}$$

The *lapse function* $N$ is introduced to normalize $n_\mu$, $n_\mu n^\mu = \varepsilon$. Moreover, the sign $\varepsilon$ in $n_\mu$ is a convenient choice for the direction of $n_\mu$ (see [44]), such that $\varepsilon^2 = 1$. We impose orthogonality for the frame decomposition (34) by virtue of

$$\tilde{\phi}^a(\varphi_b) = \delta_b^a = \varphi_b(\tilde{\phi}^a), \qquad \phi(\tilde{\varphi}) = 1 = \tilde{\varphi}(\phi), \tag{37}$$

with all other pairings between the $\phi$ and $\varphi$ frames vanishing, for instance $\phi(\varphi_a) = 0$. These pairings imply that the tensors $e_a^\mu$, $E_\mu^a$, $t^\mu$ and $n_\mu$ are not independent of each other. To see this, we insert the decompositions of frame and coframe (34) into their defining relation $\theta^\mu(\vartheta_\nu) = \delta_\nu^\mu$ to obtain

$$\delta_\nu^\mu = e_a^\mu E_\nu^a + \frac{\varepsilon}{N} t^\mu n_\nu \,. \tag{38}$$

Next, we contract this equation with $n^\mu$ as well as $e_\nu^a := \gamma^{ab} g_{\mu\nu} e_b^\mu$ to find the relations

$$t^\mu = N n^\mu + N^a e_a^\mu \quad \text{and} \quad E_\mu^a = e_\mu^a - \frac{\varepsilon}{N} n_\mu t^\nu e_\nu^a = e_\mu^a - \frac{\varepsilon}{N} n_\mu N^a - \varepsilon n_\mu n^\nu e_\nu^a, \tag{39}$$

where we define the *shift vector* $N^a := -N n^\mu E_\mu^a$. Note that the expressions (39) relate the fields $e_a^\mu$, $E_\mu^a$, $t^\mu$ and $n_\mu$ with each other. Nevertheless, their relation may be examined further by means of[8]

$$n_\mu e_a^\mu = 0 \,, \tag{40}$$

which implies that $t^\mu E_\mu^a = 0$ holds, as well. Using the orthogonality conditions (37) and (40), we may revisit the relations (38) and (39) and rewrite them as

$$\delta_\nu^\mu = e_a^\mu e_\nu^a + \varepsilon n^\mu n_\nu \,, \tag{41a}$$

$$t^\mu = N n^\mu + N^a e_a^\mu \,, \tag{41b}$$

$$E_\mu^a = e_\mu^a - \frac{\varepsilon}{N} n_\mu N^a \,. \tag{41c}$$

In the following section we use these equations to find a version of the frame decomposition (34) which is adapted to the foliation of the manifold $\mathcal{M}$ by means of hypersurfaces.

---

[8]Equation (40) can be proven by expressing the completeness relation (38) in terms of $e_a^\mu$, $n^\mu$ via (39) to find $\delta_\nu^\mu = e_a^\mu e_\nu^a + \varepsilon n^\mu n_\nu - \varepsilon n^\rho e_\rho^a e_a^\mu n_\nu$. Using this, we can write $n_\mu e_a^\mu = n_\mu \delta_\nu^\mu e_a^\nu = n_\mu e_a^\mu (\varepsilon n^\rho e_\rho^b n_\nu e_b^\nu)$. This implies $n_\mu e_a^\mu = 0$, since $1 = \varepsilon n^\rho e_\rho^a n_\nu e_a^\nu$ leads to $1 = 0$ when we take the trace of the completeness relation.

## 4.2 Adapted frame decomposition

In the frame decomposition (34) we have introduced four a priori independent tensors $e_a^\mu$, $E_\mu^a$, $t^\mu$ and $n_\mu$. We imposed orthogonality conditions in (37) and saw that they allow to express $E_\mu^a$ and $t^\mu$ in terms of $e_a^\mu$ and $n_\mu$ as in (41). Thus, we use the relations (41) to eliminate $E_\mu^a$ and $t^\mu$ in the original frame decomposition (34), to obtain

$$\theta^\mu = e_a^\mu \phi^a + N n^\mu \phi \,, \tag{42a}$$

$$\vartheta_\mu = e_\mu^a \varphi_a + \frac{\varepsilon}{N} n_\mu \varphi \,, \tag{42b}$$

where we introduced the adapted frame

$$\phi^a := \tilde{\phi}^a + N^a \phi \,, \qquad \varphi := \tilde{\varphi} - N^a \varphi_a \,. \tag{43}$$

The pairings (37) of the $(\phi^a, \phi, \varphi_a, \varphi)$ frames are invariant under this transformation and read

$$\phi^a(\varphi_b) = \delta_b^a = \varphi_b(\phi^a) \,, \qquad \phi(\varphi) = 1 = \varphi(\phi) \,, \tag{44}$$

with all remaining pairings vanishing. Hence, the new frame fields $(\phi^a, \phi, \varphi_a, \varphi)$ are orthogonal as well. To gain an intuition for what these fields are, consider a generic one-form $A = A_\mu \theta^\mu$ and a vector $B = B^\mu \vartheta_\mu$. We use (42) to expand $A$ and $B$ as

$$A = A_\mu e_a^\mu \phi^a + N A_\mu n^\mu \phi \,, \tag{45a}$$

$$B = B^\mu e_\mu^a \varphi_a + \frac{\varepsilon}{N} B^\mu n_\mu \varphi \,. \tag{45b}$$

Recall that a tensor or differential form is called tangent (normal) to a hypersurface if it vanishes when any of its indices is contracted with $n_\mu$ ($e_a^\mu$). We apply these definitions to (45) and find that tangent one-forms and vectors take the form

$$A = A_\mu e_a^\mu \phi^a \,, \qquad B = B^\mu e_\mu^a \varphi_a \,, \tag{46}$$

while normal ones are expanded as

$$A = N A_\mu n^\mu \phi \,, \qquad B = \frac{\varepsilon}{N} B^\mu n_\mu \varphi \,. \tag{47}$$

Combining these notions of being tangent and normal with the orthogonality conditions (44), we may interpret $\varphi_a$ as a frame basis on $\Sigma_\lambda$ with associated coframe basis $\phi^a$. Hence, the frame decomposition (42) is adapted to the foliation and we proceed to work with it in the following. Having set up the frame decomposition, we proceed by investigating how fields transform.

## 4.3 Decomposition of metric and connection

In the previous subsections we have investigated the decomposition of the frame fields in the foliation of the manifold $\mathcal{M}$ by means of hypersurfaces. We use this decomposition next to investigate how the basic geometric objects on $\mathcal{M}$ decompose in that foliation. In particular, apart from the frame field, we consider the metric tensor $g_{\mu\nu}$ as well as the connection one-form $\omega_\nu^\mu$ as independent dynamical fields. We start by investigating the decomposition of $g_{\mu\nu}$ by applying the adapted frame decomposition (42). Contracting (42) with $e_a^\mu$ and $n_\mu$, we exploit the normality condition (40), $n_\mu e_a^\mu = 0$, to identify $\phi = \frac{\varepsilon}{N} n_\mu \theta^\mu$ and $\phi^a = e_\mu^a \theta^\mu$. We use the manifold metric $g_{\mu\nu}$ and the hypersurface metric $\gamma_{ab} = e_a^\mu e_b^\nu g_{\mu\nu}$ to raise and lower

indices in $e_\mu^a = \gamma^{ab} g_{\mu\nu} e_b^\nu$. Using these relations, we rewrite the metric tensor by means of (42) as

$$\mathrm{d}s^2 = g_{\mu\nu}\theta^\mu \otimes \theta^\nu = \left(\gamma_{ab}e_\mu^a e_\nu^b + \varepsilon n_\mu n_\nu\right)\theta^\mu \otimes \theta^\nu. \tag{48}$$

From (48) we can read of the $3+1$ decomposition of the metric tensor $g_{\mu\nu}$ as

$$g_{\mu\nu} = \gamma_{ab}e_\mu^a e_\nu^b + \varepsilon n_\mu n_\nu, \tag{49}$$

in hypersurface tangent and normal contributions. The decomposition (49) is used heavily in the following discussion.

For deriving the transformation of the connection one-form $\omega_\nu^\mu$ to the hypersurfaces, we use the transformation law of connections with respect to the metric-affine group. For $\Lambda^\mu{}_\nu \in$ GL(n,$\mathbb{R}$), this transformation is given by [23]

$$\omega^\mu{}_\nu \to \omega'^\mu{}_\nu = \Lambda^{-1\mu}{}_\rho \omega^\rho{}_\sigma \Lambda^\sigma{}_\nu + \Lambda^{-1\mu}{}_\rho \mathrm{d}\Lambda^\rho{}_\nu. \tag{50}$$

Motivated by the frame decomposition (42) we choose the transformation

$$\begin{aligned}
\Lambda^\mu{}_\nu &:= e_a^\mu \delta_\nu^a + N n^\mu \delta_\nu^{n-1}, \\
\Lambda^{-1\mu}{}_\nu &= e_\nu^a \delta_a^\mu + \frac{\varepsilon}{N} n_\nu \delta_{n-1}^\mu.
\end{aligned} \tag{51}$$

After inserting this transformation into (50), we only consider the $\mu, \nu = (0, \ldots, n-2)$ components of $\omega'^\mu{}_\nu$ to find

$$\omega_a^b = e_\mu^b \left(\mathrm{d}e_a^\mu + \omega^\mu{}_\nu e_a^\nu\right) \qquad \Longleftrightarrow \qquad e_\mu^b D e_a^\mu = 0, \tag{52}$$

where we suppressed the prime on the left hand side since the indices immediately clarify which connection is meant here. This is the transformation of the connection on $\mathcal{M}$ to the connection on $\Sigma_\lambda$. Essentially, the above argument is a proof of the vielbein postulate for the frame field on the hypersurface for MAG, along the lines of a similar proof in [23]. We interpret the hypersurface connection coefficients by means of (52) as the contribution of $D e_a^\mu$ which is tangent to the hypersurfaces. It is, thus, natural to consider its normal contributions next to obtain an expression for $D e_a^\mu$, which constitutes a differential form version of the Gauß-Weingarten equation.

## 4.4 Foundations of the 3+1 formalism in general frames

In this section we derive the fundamental equations of the $3+1$ formalism in terms of differential forms. This involves the decomposition of $D e_a^\mu$ into contributions normal and tangent to the hypersurfaces in the foliation of the manifold $\mathcal{M}$, which can be considered as a generalized version of the Gauß-Weingarten equations. We furthermore derive a similar equation for the normal vector $n^\mu$, and take exterior covariant derivatives of these relations to derive generalized versions of the Ricci identities. This set of equations is the foundation of the $3+1$ formalism in differential forms.

We start our calculation by investigating $D e_a^\mu$. While we found the tangent contributions of this one-form for (52), we still need to calculate its hypersurface normal parts. To that end we define a one-form corresponding to the extrinsic curvature tensor $K^a{}_b := e_\mu^a e_b^\nu \nabla_\nu n^\mu$. Due to non-vanishing non-metricity, we furthermore define extrinsic curvature in a slightly different manner by $\tilde{K}_{ab} := e_a^\mu e_b^\nu \nabla_\nu n_\mu$, and we work with both definitions interchangeably in the following. Thus, we also define two different extrinsic curvature one-forms by

$$K^a := e_\mu^a D n^\mu \quad \text{and} \quad \tilde{K}_a := e_a^\mu D n_\mu. \tag{53}$$

We use the frame decomposition (42) to calculate the components of these quantities in our foliation,

$$K^a = K^a{}_b \phi^b + N e^a_\mu a^\mu \phi,$$
$$\tilde{K}_a = \tilde{K}_{ab} \phi^b + N e^\mu_a \tilde{a}_\mu \phi,$$

(54)

where we defined $a^\mu := \nabla_{\mathbf{n}} n^\mu$ and $\tilde{a}_\mu := \nabla_{\mathbf{n}} n_\mu$ analogous to the different notions of extrinsic curvature.

With the extrinsic curvature one-form at hand, we now investigate the covariant derivative of generic forms $A^\mu = e^\mu_a A^a$ tangent to $\Sigma_\lambda$. We use the $3+1$ decomposition of the metric (49) and the vielbein postulate (52) to evaluate

$$DA^\mu = D\left(g^\mu_\nu A^\nu\right) = \left(e^\mu_a e^a_\nu + \varepsilon n^\mu n_\nu\right) DA^\nu = e^\mu_a DA^a - \varepsilon n^\mu e^\nu_a Dn_\nu \wedge A^a.$$

(55)

At the same time $A^\mu$ being tangent to the hypersurface implies

$$DA^\mu = D\left(e^\mu_a A^a\right) = De^\mu_a \wedge A^a + e^\mu_a DA^a.$$

(56)

We compare both equations and noting they hold for arbitrary $A^a$, we find

$$De^\mu_a = -\varepsilon n^\mu \tilde{K}_a.$$

(57)

This is the expression of the Gauß-Weingarten equation in terms of differential forms.

We can derive a similar expression like the Gauß-Weingarten equation for the normal vector $n^\mu$. To that end, we first need to define the non-metricity one-form,

$$Q_{\mu\nu} := -Dg_{\mu\nu} = Q_{\mu\nu\rho}\theta^\rho,$$

(58)

with its tensor components given by $Q_{\mu\nu\rho} = -\nabla_\rho g_{\mu\nu}$. Abbreviating $Q_{\mathbf{nn}} := n^\mu n^\nu Q_{\mu\nu}$, we investigate $0 = D\varepsilon$ to obtain

$$n_\mu Dn^\mu = \frac{1}{2}Q_{\mathbf{nn}}.$$

(59)

We use this result for the calculation of $Dn^\mu$ and follow the same steps as described in the case of $De^\mu_a$ above to find

$$Dn^\mu = e^\mu_a K^a + \frac{\varepsilon}{2}n^\mu Q_{\mathbf{nn}}.$$

(60)

This is the hypersurface normal Gauß-Weingarten equation.

The last missing fundamental equations of the differential geometric $3+1$ decomposition are the Ricci identities for $e^\mu_a$ and $n^\mu$. Straightforward evaluation of the second covariant exterior derivatives of $e^\mu_a$ and $n^\mu$ yields

$$D^2 e^\mu_a = e^\nu_a \Omega^\mu{}_\nu - e^\mu_b \Omega^b{}_a,$$

(61a)

$$D^2 n^\mu = \Omega^\mu{}_\nu n^\nu,$$

(61b)

where the curvature two-form on $\mathcal{M}$ is given by

$$\Omega^\mu{}_\nu := \mathrm{d}\omega^\mu_\nu + \omega^\mu_\rho \wedge \omega^\rho_\nu = \frac{1}{2}R^\mu{}_{\nu\rho\sigma}\theta^\rho \wedge \theta^\sigma,$$

(62)

and we defined $\Omega^a{}_b := \mathrm{d}\omega^a_b + \omega^a_c \wedge \omega^c_b$, the boundary curvature, analogously. The tensor components $R^\mu{}_{\nu\rho\sigma}$ on the right hand side of (62) are those of the Riemann curvature tensor.

The equations we have derived in this section are sufficient to derive the Gauß-Codazzi equations, which are at the heart of the $3+1$ formalism. They describe how curvature decomposes into contributions normal and tangent to the hypersurfaces of the foliation. We perform the derivation of these equations and their analogues for torsion and non-metricity in the following subsection. We begin by investigating the $3+1$ decomposition of non-metricity.

## 4.5 Decomposition of the field strengths

In order to examine which boundary terms the non-metricity one-form $Q_{\mu\nu}$ admits, we decompose this differential form into contributions normal and tangent to the hypersurfaces given by the foliation of $\mathcal{M}$. For this derivation we make extensive use of the $3+1$ decompositions derived in the previous subsections, among which the generalized Gauß-Weingarten equations (57), (60) are particularly important. We start our investigation by using (41a) to decompose uncontracted manifold indices of a differential form $A_\mu$ as

$$A_\mu = \delta_\mu^{\ \nu} A_\nu = e_\mu^a e_a^\nu A_\nu + \varepsilon n_\mu n^\nu A_\nu. \tag{63}$$

Analogous decompositions hold for contravariant indices. We apply this index decomposition to the non-metricity one-form $Q_{\mu\nu}$ to obtain[9]

$$Q_{\mu\nu} = e_\mu^a e_\nu^b \left( e_a^\alpha e_b^\beta Q_{\alpha\beta} \right) + 2\varepsilon e_{(\mu}^a n_{\nu)} \left( e_a^\alpha n^\beta Q_{\alpha\beta} \right) + n_\mu n_\nu \left( n^\alpha n^\beta Q_{\alpha\beta} \right). \tag{64}$$

To make the decomposition (64) useful, we recall the definition of non-metricity (58) and use the $3+1$ decomposition of the metric (49) along with with repeated use of the generalized Gauß-Weingarten equation (57) and its normal counterpart (60). We obtain

$$Q_{\mu\nu} = e_\mu^a e_\nu^b (-D\gamma_{ab}) + 2\varepsilon e_{(\mu}^a n_{\nu)} \left( K_a - \tilde{K}_a \right) + 2 n_\mu n_\nu n_\alpha D n^\alpha. \tag{65}$$

This result decomposes the indices of $Q_{\mu\nu}$ into contributions normal and tangent to the hypersurfaces in the foliation of $\mathcal{M}$. In contrast to (64), (65) expresses these contributions solely in terms of fundamental fields on the hypersurface. Hence, the comparison of (65) and (64) yields the $3+1$ decomposition of the non-metricity one-form that reads

$$e_a^\mu e_b^\nu Q_{\mu\nu} = -D\gamma_{ab}, \qquad e_a^\mu n^\nu Q_{\mu\nu} = K_a - \tilde{K}_a, \qquad n^\mu n^\nu Q_{\mu\nu} = 2 n_\mu D n^\mu. \tag{66}$$

This is the decomposition of non-metricity we aim for. The projections on the left hand sides of (66) are expressed through boundary data on the right hand sides. Note that $n_\mu D n^\mu$ is a manifold scalar and does not have to be express through boundary data any further.

Next we derive the corresponding $3+1$ decompositions for torsion and curvature. Torsion is defined as the field strength of the coframe field as

$$T^\mu := D\theta^\mu = \frac{1}{2} T^\mu_{\ \rho\sigma} \theta^\rho \wedge \theta^\sigma. \tag{67}$$

As in the case of non-metricity, we need to decompose the torsion two-form into contributions which are normal and tangent to the hypersurfaces w.r.t. the foliation of the manifold. The calculation proceeds identically to that of the non-metricity decomposition above. Its result is that the torsion two-form is decomposed as

$$e_\mu^a T^\mu = D\phi^a + N K^a \wedge \phi = T^a + N K^a \wedge \phi, \tag{68a}$$

$$n_\mu T^\mu = -\tilde{K}_a \wedge \phi^a + \varepsilon D(N\phi) + \frac{N}{2} Q_{\mathbf{nn}} \wedge \phi, \tag{68b}$$

with $T^a$ the torsion 2-form on the hypersurfaces.

The final $3+1$ decomposition of the field strengths to consider is the one of the curvature two-form defined in (62). In addition to the methods used for non-metricity and torsion we

---

[9]We use symmetrization as $A_{(\mu_1\dots\mu_p)} := \frac{1}{p!} \sum_{\sigma\in\mathcal{S}} A_{\sigma(\mu_1)\dots\sigma(\mu_p)}$ over all permutations $\sigma$ in the permutation group $\mathcal{S}$. Likewise, we define $A_{[\mu_1\dots\mu_p]} := \frac{1}{p!} \sum_{\sigma\in\mathcal{S}} \mathrm{sgn}(\sigma) A_{\sigma(\mu_1)\dots\sigma(\mu_p)}$ as antisymmetrization, where $\mathrm{sgn}(\sigma)$ equals $+1$ if $\sigma$ consists of an even number of transpositions and $-1$ else.

exploit the Ricci identities (61) for $e_a^\mu$ and $n^\mu$ in the derivation of the curvature decomposition. Apart from that, the analysis proceeds similar to the one of non-metricity and yields

$$e_\mu^a e_b^\nu \Omega^\mu{}_\nu = \Omega^a{}_b - \varepsilon K^a \wedge \tilde{K}_b, \qquad n_\mu e_a^\nu \Omega^\mu{}_\nu = -D\tilde{K}_a + \frac{\varepsilon}{2}\tilde{K}_a \wedge Q_{\mathbf{nn}},$$

$$e_\mu^a n^\nu \Omega^\mu{}_\nu = DK^a + \frac{\varepsilon}{2}K^a \wedge Q_{\mathbf{nn}}, \qquad n_\mu n^\nu \Omega^\mu{}_\nu = \frac{1}{2}DQ_{\mathbf{nn}} + K^a \wedge \tilde{K}_a. \tag{69}$$

These equations are generalizations of the Gauß-Codazzi equations. As we have seen in section 3, they are particularly relevant for the calculation of GHY terms if a specific Lagrangian $\mathcal{L}$ is considered. In particular, the calculation of GHY terms which are of second or higher order in curvature always involves terms like $e_\mu^a n^\nu \Omega^\mu{}_\nu$ which are evaluated by means of (69). Moreover, as we show next, the $3+1$ decompositions of curvature, torsion and non-metricity which we have just derived are sufficient for calculating the GHY term for generic Lagrangians which are polynomials or Taylor series of these field strengths.

### 4.6 Post-Riemannian Gibbons-Hawking-York term

This section follows [46] to derive a generalized GHY term from the foliations of curvature, torsion and non-metricity. For the sake of generality we consider an action of the form

$$S_{\text{orig}}[g_{\mu\nu}, \omega_\nu^\mu, \theta^\mu] = \int_{\mathcal{M}} \mathcal{L}(\Omega^\mu{}_\nu, T^\mu, Q_{\mu\nu}), \tag{70}$$

where the Lagrangian $\mathcal{L}$ is a polynomial in curvature $\Omega^\mu{}_\nu$, torsion $T^\mu$ and non-metricity $Q_{\mu\nu}$.[10] We introduce Lagrange multipliers $\varphi_\mu{}^\nu$, $t_\mu$ and $q^{\mu\nu}$ in order to linearize $\mathcal{L}$ and derive the GHY term more easily.

$$S[g_{\mu\nu}, \omega_\nu^\mu, \theta^\mu, \varphi_\mu{}^\nu, \varrho^\mu{}_\nu, t_\mu, \tau^\mu, q^{\mu\nu}, \sigma_{\mu\nu}] = \int_{\mathcal{M}} \Big[ \mathcal{L}(\varrho^\mu{}_\nu, \tau^\mu, \sigma_{\mu\nu}) + *\varphi_\mu{}^\nu \wedge (\Omega^\mu{}_\nu - \varrho^\mu{}_\nu) \quad (71)$$

$$+ *t_\mu \wedge (T^\mu - \tau^\mu) + *q^{\mu\nu} \wedge (Q_{\mu\nu} - \sigma_{\mu\nu}) \Big].$$

Integrating out the multipliers shows that the action (71) yields the same equations of motion as (70). The remaining fields $\varrho^\mu{}_\nu$, $\tau^\mu$ and $\sigma_{\mu\nu}$ in (71) introduced are auxiliary fields that we demand to have the same symmetries as the corresponding field strength. This symmetry identification is also required for the Lagrange multipliers $\varphi_\mu{}^\nu$, $t_\mu$ and $q^{\mu\nu}$ in order to allow equivalence of the equations of motion of (70) and (71) [46].

To proceed we substitute the $3+1$ decompositions of $\Omega^\mu{}_\nu$, $T^\mu$ and $Q_{\mu\nu}$ in the action (71) in order to decompose $S$ into boundary tangent and normal contributions and, subsequently, isolate the boundary terms. For this, we write by means of (41a) for the linearized terms in $S$

$$*\varphi_\mu{}^\nu \wedge \Omega^\mu{}_\nu = \left(e_a^\alpha e_\beta^b * \varphi_\alpha{}^\beta\right) \wedge \left(e_\mu^a e_b^\nu \Omega^\mu{}_\nu\right) + \varepsilon\left(e_a^\alpha n_\beta * \varphi_\alpha{}^\beta\right) \wedge \left(e_\mu^a n^\nu \Omega^\mu{}_\nu\right)$$

$$+ \varepsilon\left(n^\alpha e_\beta^a * \varphi_\alpha{}^\beta\right) \wedge \left(n_\mu e_a^\nu \Omega^\mu{}_\nu\right) + \left(n^\alpha n_\beta * \varphi_\alpha{}^\beta\right) \wedge \left(n_\mu n^\nu \Omega^\mu{}_\nu\right), \tag{72a}$$

$$*t_\mu \wedge T^\mu = \left(e_a^\nu * t_\nu\right) \wedge \left(e_\mu^a T^\mu\right) + \varepsilon\left(n^\nu * t_\nu\right) \wedge \left(n_\mu T^\mu\right), \tag{72b}$$

$$*q^{\mu\nu} \wedge Q_{\mu\nu} = \left(e_\alpha^a e_\beta^b * q^{\alpha\beta}\right) \wedge \left(e_a^\mu e_b^\nu Q_{\mu\nu}\right) + \varepsilon\left(e_\alpha^a n_\beta * q^{\alpha\beta}\right) \wedge \left(e_a^\mu n^\nu Q_{\mu\nu}\right)$$

$$+ \varepsilon\left(n_\alpha e_\beta^a * q^{\alpha\beta}\right) \wedge \left(n^\mu e_a^\nu Q_{\mu\nu}\right) + \left(n_\alpha n_\beta * q^{\alpha\beta}\right) \wedge \left(n^\mu n^\nu Q_{\mu\nu}\right). \tag{72c}$$

---

[10]Derivatives of $\Omega^\mu{}_\nu$, $T^\mu$ or $Q_{\mu\nu}$ may be reduced to polynomials by means of the Bianchi identities (2).

The right hand side of each of these equations can be written in terms of the $3+1$ decompositions of $Q_{\mu\nu}$, $T^\mu$ and $\Omega^\mu{}_\nu$ shown in (66), (68) and (69). For example, (72a) takes the form

$$
*\varphi_\mu{}^\nu \wedge \Omega^\mu{}_\nu = \left(e_a^\alpha e_\beta^b * \varphi_\alpha{}^\beta\right) \wedge \left(\Omega^a{}_b - \varepsilon K^a \wedge \tilde{K}_b\right) + \varepsilon\left(e_a^\alpha n_\beta * \varphi_\alpha{}^\beta\right) \wedge \left(DK^a + \frac{\varepsilon}{2}K^a \wedge Q_{\mathbf{nn}}\right)
$$
$$
+ \varepsilon\left(n^\alpha e_\beta^a * \varphi_\alpha{}^\beta\right) \wedge \left(-D\tilde{K}_a + \frac{\varepsilon}{2}\tilde{K}_a \wedge Q_{\mathbf{nn}}\right) + \left(n^\alpha n_\beta * \varphi_\alpha{}^\beta\right) \wedge \left(\frac{1}{2}DQ_{\mathbf{nn}} + K^a \wedge \tilde{K}_a\right).
$$
$$(73)$$

We insert this decomposition and the analogues for torsion and non-metricity into the action (71). The derivative terms yield by means of Stokes' theorem [58, 59] a boundary action, which reads

$$
S_{\mathrm{bdy}} = \int_{\partial\mathcal{M}} \left[\omega^a_b \wedge \left(e_a^\alpha e_\beta^b * \varphi_\alpha{}^\beta\right) + \phi^a \wedge \left(e_a^\nu * t_\nu\right) - \gamma_{ab}\left(e_\alpha^a e_\beta^b * q^{\alpha\beta}\right)\right.
$$
$$
\left.\left. - \varepsilon\tilde{K}_a \wedge \left(n^\alpha e_\beta^a * \varphi_\alpha{}^\beta\right) + \varepsilon K^a \wedge \left(e_a^\alpha n_\beta * \varphi_\alpha{}^\beta\right) + \frac{1}{2}Q_{\mathbf{nn}} \wedge \left(n^\alpha n_\beta * \varphi_\alpha{}^\beta\right)\right]\right|_{\partial\mathcal{M}}.
$$
$$(74)$$

Note that we use the restriction of the integrand in (74) to $\partial\mathcal{M}$ as abbreviation for its pullback to $\partial\mathcal{M}$.

We finally construct the GHY term which cancels the boundary contributions to the action (71). Only some of the terms in (74) actually contribute boundary terms when we consider variations of the action. In particular, the variational principle assumes that $\delta g_{\mu\nu}|_{\partial\mathcal{M}} = 0$, $\delta\theta^\mu|_{\partial\mathcal{M}} = 0$ and $\delta\omega^\mu_\nu|_{\partial\mathcal{M}} = 0$. According to [38] it equivalently suffices to demand that the Dirichlet boundary conditions $\delta\gamma_{ab} = 0$, $\delta\phi^a = 0$ and $\delta\omega^a_b = 0$ hold since the original conditions may be reinstated by gauge transformations on $\partial\mathcal{M}$.[11] This implies that the GHY term of (70) only has to cancel the contributions from the second line of (74) and, thus, takes the form

$$
S_{\mathrm{GHY}} = -\int_{\partial\mathcal{M}} \left(-\varepsilon\tilde{K}_a \wedge *\varphi^{\mathbf{n}a} + \varepsilon K^a \wedge *\varphi_{a\mathbf{n}} + Q_{\mathbf{nn}} \wedge \frac{1}{2} *\varphi_{\mathbf{nn}}\right)\bigg|_{\partial\mathcal{M}},
\qquad (75)
$$

where we abbreviated $*\varphi^{\mathbf{n}a} \equiv n^\alpha e_\beta^a * \varphi_\alpha{}^\beta$, $*\varphi_{a\mathbf{n}} \equiv e_a^\alpha n_\beta * \varphi_\alpha{}^\beta$ and $*\varphi_{\mathbf{nn}} \equiv n^\alpha n_\beta * \varphi_\alpha{}^\beta$. The result (75) is the general form of the GHY term in theories with torsion and non-metricity we aim for. It is the main result of this paper. In particular, (75) is the GHY term for any theory which is constituted by a Lagrangian $\mathcal{L}$ that is, possibly an infinite, polynomial in curvature, torsion and non-metricity.

To apply our result to a specific Lagrangian $\mathcal{L}$ in (70), we need to know the Lagrange multipliers $\varphi^{\mathbf{n}a}$, $\varphi_{a\mathbf{n}}$ and $\varphi_{\mathbf{nn}}$. To obtain these, we write the action (71) in a $3+1$ form in terms of $(\varphi^{\mathbf{n}a}, \varphi_{a\mathbf{n}}, \varphi_{\mathbf{nn}}, \dots)$ instead of $\varphi_\mu{}^\nu$, using (72). Then we can read the required components of $\varphi$ from the equations of motion of $\varrho_{\mu\nu}$

$$
\begin{aligned}
*\varphi^{\mathbf{n}a} \wedge \delta\varrho_{\mathbf{n}a} &= \varepsilon\delta_{\varrho_{\mathbf{n}a}}\mathcal{L}(\varrho_{\mathbf{n}a}, \varrho^{a\mathbf{n}}, \varrho_{\mathbf{nn}}, \dots), \\
*\varphi_{a\mathbf{n}} \wedge \delta\varrho^{a\mathbf{n}} &= \varepsilon\delta_{\varrho^{a\mathbf{n}}}\mathcal{L}(\varrho_{\mathbf{n}a}, \varrho^{a\mathbf{n}}, \varrho_{\mathbf{nn}}, \dots), \\
*\varphi^{\mathbf{nn}} \wedge \delta\varrho_{\mathbf{nn}} &= \delta_{\varrho_{\mathbf{nn}}}\mathcal{L}(\varrho_{\mathbf{n}a}, \varrho^{a\mathbf{n}}, \varrho_{\mathbf{nn}}, \dots),
\end{aligned}
\qquad (76)
$$

derived from (71). Through these constraints, we may calculate $S_{\mathrm{GHY}}$ for *any* given action polynomial in curvature, torsion and non-metricity.

---

[11]By the same argument the residual terms of the curvature, torsion and non-metricity foliations which are not exact forms do not contribute to the GHY term. We have, hence, disregarded them in the consideration above.

We emphasize that (75) and (76) are the only equations necessary to calculate the GHY term for a specific action. The advantage of our method is, thus, not only the universality of our results but furthermore the efficiency of the calculation. We have already observed this efficiency in the examples examined in section 3.

Finally we note that the GHY term (75) simplifies considerably if non-metricity is absent as shown below.

### 4.6.1 Gibbons-Hawking-York term for metric compatible theories

Let us simplify the above result for theories which are metric compatible, that is, in which the non-metricity one-form $Q_{\mu\nu}$ vanishes. For metric compatible theories, the Bianchi identity of non-metricity (2), $\Omega_{\mu\nu} + \Omega_{\nu\mu} = DQ_{\mu\nu}$, forces the curvature two-form to be antisymmetric. Since the corresponding Lagrange multipliers $\varphi$ and $\varrho$ are required to have the same symmetry as $\Omega^{\mu}{}_{\nu}$, it follows that $\varphi$ is antisymmetric in its two indices. This allows to simplify the Gibbons-Hawking York term as

$$S_{\text{GHY}}^{Q=0} = 2 \int_{\partial\mathcal{M}} \varepsilon K^a \wedge *\varphi_{\mathbf{n}a}|_{\partial\mathcal{M}} \tag{77}$$

in the metric-compatible case. Note that due to the antisymmetry of the curvature two-form, formally we must consider the constraints (76) as not independent. Nevertheless, we may forego this caveat if we treat $\delta\varrho_{\mathbf{n}a}$ and $\delta\varrho_{a\mathbf{n}}$ as independent variations and use the symmetry conditions only after the variational calculus. In appendix B we show explicitly that both methods yield the same result. This concludes the explicit derivation of the GHY boundary term for any action polynomial in curvature, torsion and non-metricity, as anticipated in section 2. We refer to the interpretation of our results in section 2.

## 5 Summary and discussion

In conclusion, in this paper we have accomplished the first step towards the full holographic renormalization of metric-affine gravity (MAG) theories, namely the derivation of the Gibbons-Hawking-York term (12) in closed form for any given polynomial action (5) involving curvature, torsion and non-metricity.[12] In addition, our method for calculating the GHY term is very efficient in practical terms, since it amounts to evaluating a single variation, (10). Interpreting the main result (12), we notice that only explicitly curvature-related terms in the Lagrangian contribute to the GHY boundary terms. If a Lagrangian depends solely on torsion and non-metricity, we do not need to introduce GHY terms in order for the Dirichlet variational problem to be well-defined.[13]

We tested our method explicitly for two theories with known GHY terms, Einstein-Hilbert gravity and 4d Chern-Simons modified gravity, in sections 3.1 and 3.2 respectively. Moreover, we used our method to derive the GHY term for torsionful Lovelock-Chern-Simons theory in section 3.3 and for metric-affine gravity in appendix A. Our new result for the Lovelock-Chern-Simons GHY term meets intuitive expectations: Since the Lovelock-Chern-Simons action has a form similar to the Einstein-Hilbert and 4d Chern-Simons modified ones, we expect their GHY terms to also be comparable. This expectation bears out as explained in more detail in section 3.3. We emphasize that for Lovelock-Chern-Simons theory, our result generalizes that of [16] to off-shell field configurations and that of [52] to torsionful backgrounds. It would

---

[12]This includes non-polynomial Lagrangians that can be Taylor expanded in the field strengths.

[13]Of course, boundary terms may arise when these theories must satisfy additional constraints, such as setting the curvature two-form $\Omega^{\mu}{}_{\nu} = 0$ as in teleparallel theories of gravity [47, 48].

be interesting to evaluate our GHY term on the solution of [16] and to generalize the method of [52] to torsionful backgrounds as a consistency check of our formalism.

The next step towards understanding spin and hypermomentum transport by means of the AdS/CFT correspondence is now to complete the holographic renormalization procedure for MAG theories.[14] Evaluating the MAG action and the associated GHY term given in appendix A on the torsionful and non-metric AdS Reissner-Nordström solution [60] will provide a hint on what kind of divergences may appear in this procedure. In general, it will be necessary to find the asymptotic expansion for torsion and non-metricity coupled to an asymptotically AdS metric. This will provide us with the holographic dictionary for torsion and non-metricity. We then have to follow e.g. [61] and calculate the regularized on-shell action in order to find all possible divergences, and to construct appropriate counterterms to cancel them. This will allow us to derive the thermodynamic properties of black holes with spin and hypermomentum. It may also enable us to study Hawking-Page type phase transitions [62] relying entirely on spin and hypermomentum. In fact we expect such transitions to exist, since the entropy of a black hole generically decreases when it starts rotating [63]. Holographic renormalization then sets the stage for applying the fluid/gravity correspondence [42, 64] to MAG and for deriving the complete hydrodynamic expansion of strongly-coupled holographic systems with non-trivial spin and hypermomentum transport. We note that non-trivial spin transport in hydrodynamics has already been discussed in [15–21], but hypermomentum transport is still mostly unexplored. The inclusion of hypermomentum is particularly interesting, since it contains degrees of freedom largely left unexplored in the literature.[15] These degrees of freedom are expected to lead to interesting changes in the transport properties of matter, since their source, non-metricity, is interpreted geometrically as the modification of the causal structure of spacetime. Since non-metricity is by definition first order in the hydrodynamic derivative expansion, we expect it to contribute to hydrodynamic transport only at second order in derivatives or higher. This makes conformal fluids, where the second order derivative expansion is well-known [66], the most suitable candidate for carrying out this procedure.

Apart from the AdS/CFT correspondence, MAG theories are relevant in their own right. For example, they provide us with extensions of Einstein's theory of relativity based on a gauge principle [23]. Thus, they are a promising standing point for exploring deviations from general relativity in search for clues and constraints helpful for the construction of consistent quantum gravity models. Moreover, torsion appears naturally in supergravity and, hence, top-down string theory constructions [67]. In addition, non-metricity may play a role for conformal gravity as follows: When the non-metricity is pure trace, i.e. $Q_{\mu\nu} = \sigma g_{\mu\nu}$, it may be absorbed into the covariant derivative. This derivative is then metric-compatible and covariant under both diffeomorphisms and Weyl transformations (see e.g. [65]). We expect such derivatives to be useful for the construction of non-local actions such as the Riegert action that gives rise to the Euler term in the four-dimensional conformal anomaly [68], and for obtaining Weyl covariant differential operators [69, 70] and curvature scalars [71, 72]. Finally, restricting ourselves to the realm of classical gravity, our results may be applied to MAG theories to test cosmological implications of deviations from Einstein gravity along the lines suggested in [73].

To conclude, we note that our results may also be useful within the context of studying topological field theories in the presence of torsion and non-metricity. Topological field theories constitute a convenient way of deriving non-dissipative transport properties, in this case for spin and hypermomentum, without the necessity of solving the underlying electronic dynamics (see [7] for instance). For example, we may follow [74] instead of a more complicated entropy current analysis. Our results and specifically our hypersurface decomposition for torsion and non-metricity allows writing down such theories in a spacetime with a timelike boundary. We

---

[14]Recall spin and hypermomentum are dual to torsion and non-metricity respectively [23].

[15]See [65] for a discussion on non-metricity of the Weyl type.

expect them to be relevant for describing systems relevant for condensed matter physics.

# 6 Acknowledgments

We thank Daniel Grumiller, Umut Gürsoy, Stefan Theisen, and Zhou-Yu Xian for useful discussions. The work in Würzburg is funded by the Deutsche Forschungsgemeinschaft (DFG, German Research Foundation) through Project-ID 258499086—SFB 1170 'ToCoTronics' and through the Würzburg-Dresden Cluster of Excellence on Complexity and Topology in Quantum Matter - ct.qmat Project-ID 390858490—EXC 2147, and in part through the German-Israeli Project Cooperation (DIP) grant "Holography and the Swampland". The research of B.H. is funded by the Bundesministerium für Bildung und Forschung (BMBF, German Federal Ministry of Education and Research) through the Cusanuswerk - Bischöfliche Studienförderung. I.M.'s research is supported by the 'Curiosity Driven Grant 2020' of the University of Genova and by the INFN Scientific Initiatives SFT: 'Statistical Field Theory, Low-Dimensional Systems, Integrable Models and Applications'.

# A GHY term for metric-affine gravity (MAG)

Metric-affine gravity contains non-vanishing non-metricity in addition to curvature and torsion. According to [23,60] the most general Lagrangian density which is at most quadratic in these fields and parity conserving is

$$
\begin{aligned}
V_{\text{MAG}} = \frac{1}{2\kappa} \Bigg[ & -a_0 R^{\alpha\beta} \wedge \eta_{\alpha\beta} - 2\Lambda\eta + T^\alpha \wedge * \left( \sum_{I=1}^{3} a_I \, ^{(I)}T_\alpha \right) \\
& + 2 \left( \sum_{I=2}^{4} c_I \, ^{(I)}Q_{\alpha\beta} \right) \wedge \theta^\alpha \wedge *T^\beta + Q_{\alpha\beta} \wedge * \left( \sum_{I=1}^{4} b_I \, ^{(I)}Q^{\alpha\beta} \right) \\
& + b_5 \left( ^{(3)}Q_{\alpha\gamma} \wedge \theta^\alpha \right) \wedge * \left( ^{(4)}Q^{\beta\gamma} \wedge \theta_\beta \right) \Bigg] \\
& - \frac{1}{2\rho} R^{\alpha\beta} \wedge * \Bigg[ \sum_{I=1}^{6} w_I \, ^{(I)}W_{\alpha\beta} + w_7 \theta_\alpha \wedge (e_\gamma \rfloor ^{(5)}W^\gamma{}_\beta) \\
& + \sum_{I=1}^{5} z_I \, ^{(I)}Z_{\alpha\beta} + z_6 \theta_\gamma \wedge (e_\alpha \rfloor ^{(2)}Z^\gamma{}_\beta) + \sum_{I=7}^{9} z_I \theta_\alpha \wedge (e_\gamma \rfloor ^{(I-4)}Z^\gamma{}_\beta) \Bigg] ,
\end{aligned}
\tag{A.1}
$$

where $a_0, \ldots, a_3, b_1, \ldots, b_5, c_2, c_3, c_4, w_1, \ldots, w_7, z_1, \ldots, z_9$ are dimensionless constants and the curvature two-form $R_\nu{}^\mu$ relates to the definition in (62) as $R_\nu{}^\mu = \Omega^\mu{}_\nu$. The various terms account for the 4+3+11 irreducible contributions to $Q_{\alpha\beta}$, $T^\alpha$ and $R_\alpha{}^\beta$ under the (pseudo)orthogonal group, respectively. The explicit form of these irreducible decompositions is given in appendix B of [23]. $e_\mu$ denotes the basis dual to $\theta^\mu$ and $\rfloor$ is the interior product so that $e_\mu \rfloor \theta^\nu = \delta^\nu_\mu$. While $\kappa$ is related to Newton's constant $G_n$ in $n$ dimensions as $\kappa = 8\pi G_n$, the coefficient $\rho$ controls the curvature squared terms and is, thus, called strong gravity coupling constant.

Let us consider the terms in (A.1) which are proportional to $(2\kappa)^{-1}$ first. The first one of them, $-a_0 R^{\alpha\beta} \wedge \eta_{\alpha\beta}$, is exactly the Einstein-Hilbert Lagrangian which we already discussed in section 3.1. The remainder of the terms proportional to $(2\kappa)^{-1}$ does not contain curvature and therefore does not contribute to the GHY term as we already noticed in sections 2 and 4.

Therefore, the interesting part of the MAG Lagrangian (A.1) in view of the GHY term is

$$
\begin{aligned}
V_{\text{MAG},\rho} = -\frac{1}{2\rho} R^{\alpha\beta} \wedge * \Bigg[ & \sum_{I=1}^{6} w_I {}^{(I)}W_{\alpha\beta} + w_7 \theta_\alpha \wedge \left( e_\gamma \rfloor {}^{(5)}W^\gamma{}_\beta \right) \\
& + \sum_{I=1}^{5} z_I {}^{(I)}Z_{\alpha\beta} + z_6 \theta_\gamma \wedge \left( e_\alpha \rfloor {}^{(2)}Z^\gamma{}_\beta \right) + \sum_{I=7}^{9} z_I \theta_\alpha \wedge \left( e_\gamma \rfloor {}^{(I-4)}Z^\gamma{}_\beta \right) \Bigg].
\end{aligned}
\tag{A.2}
$$

Since the number of terms in this action is large compared to the actions considered in 2, some preliminary considerations are in place before we calculate its GHY term.

## A.1 Variation of Hodge stars and interior products

For deriving the GHY term for the MAG Lagrangian we need to insert the explicit forms of the Lagrange multipliers $\varphi^{\mathbf{n}a}$, $\varphi_{a\mathbf{n}}$ and $\varphi_{\mathbf{nn}}$ into the generic GHY result (75). We derive these explicit expressions using the constraints (76) which relate $\varphi^{\mu\nu}$ to the variation of the Lagrangian with respect to $\varrho_{\mu\nu}$. Considering the relevant part of the MAG Lagrangian (A.2) it is immediately obvious that its variation involves variations of Hodge duals as well as interior products. Therefore, we investigate variations of these objects next.

To that effect, we consider differential forms $A$, $B$ of the same degree $p$ on an $n$-manifold. For evaluating expressions like $\delta * A \wedge B$ we use the relation [75–77]

$$
*A \wedge B = *B \wedge A.
\tag{A.3}
$$

Furthermore, we employ the result

$$
(\delta * - * \delta)A = \delta\theta^\alpha \wedge (e_\alpha \rfloor * A) - *[\delta\theta^\alpha \wedge (e_\alpha \rfloor A)] + \delta g_{\alpha\beta} \left[ \theta^{(\alpha} \wedge (e^{\beta)} \rfloor * A) - \frac{1}{2} g^{\alpha\beta} * A \right]
\tag{A.4}
$$

of [76] for commuting $\delta$ and $*$. The right hand side of (A.4) does not contribute at the boundary in the variational principle. Hence, it comes in handy to define $\simeq$ as being equivalent at the boundary and omitting terms which are irrelevant there. (A.4) therefore implies

$$
\delta * A \simeq * \delta A,
\tag{A.5}
$$

which we combine with (A.3) to find

$$
\delta * A \wedge B \simeq *B \wedge \delta A.
\tag{A.6}
$$

For the further evaluation of variations in combination with the Hodge duality and the interior product we use the relations [75–77]

$$
e_\alpha \rfloor A = (-1)^{n(p-1)+\text{ind}\,g} * (\theta_\alpha \wedge *A),
\tag{A.7a}
$$

$$
*(e_\alpha \rfloor A) = (-1)^{p-1} \theta_\alpha \wedge *A,
\tag{A.7b}
$$

$$
e_\alpha \rfloor * A = *(A \wedge \theta_\alpha),
\tag{A.7c}
$$

$$
**A = (-1)^{p(n-p)+\text{ind}\,g} A,
\tag{A.7d}
$$

where $\text{ind}\,g$ is the number of negative signs in the signature of $g$. The combination of (A.5) with (A.7a) immediately implies

$$
\delta \left( e_\mu \rfloor A \right) \simeq e_\mu \rfloor \delta A,
\tag{A.8}
$$

for the variation of interior products. These are all the expressions for the variation of Hodge duals and interior product which we need to calculate the variations of the terms in the GHY relevant part of the MAG Lagrangian (A.2). Therefore, we proceed by considering variations of the involved terms separately.

## A.2 Variation of the MAG Lagrangian

As first step of the irreducible decomposition of curvature we follow [23] and decompose the symmetric and antisymmetric parts of the curvature two-form as

$$R_{\mu\nu} = W_{\mu\nu} + Z_{\mu\nu}, \tag{A.9a}$$

$$\text{where} \quad W_{\mu\nu} := R_{[\mu\nu]}, \quad Z_{\mu\nu} := R_{(\mu\nu)}. \tag{A.9b}$$

From this decomposition we immediately read off

$$\delta R_{\mu\nu} = \delta W_{\mu\nu} + \delta Z_{\mu\nu}, \tag{A.10}$$

which we use to consider the variations of $W$ and $Z$ instead of the $R$ variation.

We start our investigation by considering variations of the terms in (A.2) which include the antisymmetric part of curvature, $W_{\mu\nu} = R_{[\mu\nu]}$. Performing the variations of the irreducible decompositions of $W_{\mu\nu}$ using the methods from the previous subsections we obtain

$$R^{\alpha\beta} \wedge \delta *^{(I)}W_{\alpha\beta} \simeq \delta W^{\alpha\beta} \wedge *^{(I)}W_{\alpha\beta}, \tag{A.11}$$

for all $I \in \{1, 2, \ldots, 6\}$, where $\simeq$ omits terms irrelevant for the GHY term calculation. (A.11) is a non-trivial result as we observe in the case of the term with coefficient $w_7$. For this term we obtain

$$R^{\alpha\beta} \wedge \delta * \left( \theta_\alpha \wedge \left( e_\gamma \rfloor^{(5)}W^\gamma_{\ \beta} \right) \right) \simeq -\frac{(-1)^{n+\mathrm{ind}\,g}}{n-2} \delta W_{\mu\nu} \wedge \Theta^{\nu\mu}_{\ \ [\beta\gamma]}{}^\gamma_{\ \alpha} \left( *R^{\alpha\beta} \right). \tag{A.12}$$

The operator $\Theta$ is introduced such that for any differential form $A$ we have

$$\Theta_\mu(A) := * \left( \theta_\mu \wedge A \right) \qquad \text{and} \qquad \Theta_{\mu_p \ldots \mu_1}(A) := * \theta_{\mu_p} \wedge \Theta_{\mu_{p-1} \ldots \mu_1}(A). \tag{A.13}$$

This definition fulfills $\Theta_{\mu_p \ldots \mu_1} \left( \Theta_{\nu_q \ldots \nu_1}(A) \right) = \Theta_{\mu_p \ldots \mu_1 \nu_q \ldots \nu_1}(A)$ which we use in the variation of the symmetric curvature contributions that we consider next.

We calculate the relevant terms of the $Z_{\mu\nu}$ variation like for the $W_{\mu\nu}$ one. In contrast to the latter each irreducible component of $Z_{\mu\nu}$ appears in two terms in the Lagrangian (A.2). Before we give the results of the variation we introduce the decomposition $Z_{\alpha\beta} = \slashed{Z}_{\alpha\beta} + \frac{1}{n} g_{\alpha\beta} Z$, where $Z := Z^\alpha_{\ \alpha}$ is the curvature trace.

As in the case of $W_{\mu\nu}$ some of the variations simplify significantly as

$$R^{\alpha\beta} \wedge \delta *^{(I)}Z_{\alpha\beta} \simeq \delta \slashed{Z}^{\alpha\beta} \wedge *^{(I)}Z_{\alpha\beta}, \tag{A.14}$$

for $I \in \{2, 4, 5\}$. Prominently, $I = 3$ is missing in this list. For the according term we obtain

$$R^{\alpha\beta} \wedge \delta *^{(3)}Z_{\alpha\beta} \simeq \frac{(-1)^{n+\mathrm{ind}\,g}}{n^2-4} \delta \slashed{Z}_{\mu\nu} \wedge \Theta^{\nu\mu} \left( * \left( n(n-2)\Delta - 2Z \right) \right). \tag{A.15}$$

This result does not simplify in the same way as the other terms did since the combination of terms in $^{(3)}Z_{\alpha\beta}$ is not trivial. In particular, one of the terms involved in $^{(3)}Z_{\alpha\beta}$ is proportional to $g_{\alpha\beta}$ and in the contraction with $R^{\alpha\beta}$ thus yields the factor of $Z \equiv Z^\alpha_{\ \alpha}$ which we observe in the result. But for casting the result in the way one naively expects from comparison with (A.14), (A.15) may solely depend on $\slashed{Z}_{\alpha\beta}$ but not on $Z$. Hence, the structure of $^{(3)}Z_{\alpha\beta}$ makes it impossible to rewrite (A.15) in the same way as the remaining terms.

Accordingly, the variation of $^{(1)}Z_{\alpha\beta}$ is non-trivial, too, and needs to be evaluated according to

$$R^{\alpha\beta} \wedge \delta *^{(1)}Z_{\alpha\beta} = R^{\alpha\beta} \wedge \left[ \delta * Z_{\alpha\beta} - \sum_{I=2}^{5} \delta *^{(I)}Z_{\alpha\beta} \right]. \tag{A.16}$$

The rest of the combinations of curvature terms in the MAG action mixes not only $\overset{\rightarrow}{Z}_{\alpha\beta}$ and $Z$ but furthermore $Z_{\alpha\beta}$ and $W_{\alpha\beta}$ terms in general. Hence, there is no simplification like the ones presented above. The results for these variations are

$$R^{\alpha\beta} \wedge \delta * \left( \theta^{\gamma} \wedge \left(e_{\alpha}\rfloor^{(2)}Z_{\gamma\beta}\right)\right) \simeq \frac{1}{2}\delta\overset{\rightarrow}{Z}_{\mu\nu} \wedge \theta^{\nu} \wedge \left[-(-1)^{\text{ind}g} * \left(\theta_{\alpha} \wedge \theta_{\gamma} \wedge * \left(\theta^{(\gamma|} \wedge *R^{\alpha|\mu)}\right)\right)\right.$$
$$\left. + \frac{1}{n-2}\Theta^{\mu}{}_{\beta}\left(*\left(\theta_{\alpha} \wedge \theta_{\gamma} \wedge *\left(\theta^{(\gamma|} \wedge *R^{\alpha|\beta)}\right)\right)\right)\right],$$
(A.17a)

$$R^{\alpha\beta} \wedge \delta * \left(\theta_{\alpha} \wedge \left(e^{\gamma}\rfloor^{(3)}Z_{\gamma\beta}\right)\right) \simeq \frac{1}{n^2-4}\delta\overset{\rightarrow}{Z}_{\mu\nu} \wedge \Theta^{\mu\nu}{}_{\beta}\left[n(-1)^{n+\text{ind}g}\Theta_{\gamma}{}^{(\gamma|}{}_{\alpha}\left(*R^{\alpha|\beta)}\right) - 2\Theta_{\alpha}\left(*R^{\alpha\beta}\right)\right],$$
(A.17b)

$$R^{\alpha\beta} \wedge \delta * \left(\theta_{\alpha} \wedge \left(e^{\gamma}\rfloor^{(4)}Z_{\gamma\beta}\right)\right) \simeq \frac{(-1)^{n+\text{ind}g}}{n}\delta Z \wedge \Theta_{\beta\alpha}\left(*R^{\alpha\beta}\right),$$
(A.17c)

$$R^{\alpha\beta} \wedge \delta * \left(\theta_{\alpha} \wedge \left(e^{\gamma}\rfloor^{(5)}Z_{\gamma\beta}\right)\right) \simeq \frac{(-1)^{n}}{n}\delta\overset{\rightarrow}{Z}_{\mu\nu} \wedge \Theta^{\nu}\left[2\Theta_{\gamma}{}^{(\gamma|}{}_{\alpha}\left(*R^{\alpha|\mu)}\right) + \Theta^{\mu}{}_{\beta\gamma}{}^{(\gamma|}{}_{\alpha}\left(*R^{\alpha|\beta)}\right)\right].$$
(A.17d)

For the construction of the full GHY term of MAG we need to proceed as in the examples in section 3. In particular, with the variations above it is straightforward to derive the required Lagrange multipliers from (10) and apply the Gauß-Codazzi equations (11) to simplify the remaining curvature forms in these Lagrange multipliers. Subsequently, the multipliers simplified that way need to be inserted into (12) to obtain the GHY term for MAG. We outline how this calculation is performed using the variations in (A.11- A.17) for vanishing non-metricity next.

## A.3 GHY term for metric compatible MAG

For constructing the GHY term for MAG let us consider the Lagrangian (A.2) again. For simplicity we only consider the case of vanishing non-metricity here, $Q_{\mu\nu} \equiv -Dg_{\mu\nu} = 0$. From the relation $\Omega_{\mu\nu} + \Omega_{\nu\mu} = DQ_{\mu\nu}$ for the curvature two-form $\Omega^{\mu}{}_{\nu}$ we obtain that $\Omega_{(\mu\nu)} = 0$ in this case. Comparison with the curvature decomposition in (A.9) implies that $Z_{\mu\nu} = 0$. Thus, we are left with the relevant part of the MAG Lagrangian (A.2) for the calculation of the metric compatible GHY term to be

$$V^{Q=0}_{\text{MAG},\rho} = -\frac{1}{2\rho}R^{\alpha\beta} \wedge * \left[\sum_{I=1}^{6} w_I \, {}^{(I)}W_{\alpha\beta} + w_7 \theta_{\alpha} \wedge \left(e_{\gamma}\rfloor^{(5)}W^{\gamma}{}_{\beta}\right)\right].$$
(A.18)

This is the part of the MAG Lagrangian (A.1) in the presence of torsion and vanishing non-metricity which contributes new terms to the GHY term. Hence, we use (A.18) next to determine the GHY term for metric compatible MAG with the method presented in chapter 2.

This method requires us to calculate the Lagrange multiplier $*\varphi^{\mathbf{n}a}$ first, since it is the only term which contributes to the metric compatible GHY term (13). We calculate $*\varphi^{\mathbf{n}a}$ by means of the constraints (10) to obtain

$$*\varphi_{\mathbf{n}a}|_{\varrho=R} = -\frac{1}{2\rho}\left[2\sum_{I=1}^{6}w_I * {}^{(I)}W_{\mathbf{n}a} + w_7(-1)^{n+\text{ind}g}\Theta_{[\mathbf{n}|\gamma}\left(*^{(5)}W^{\gamma}{}_{|a]}\right)\right.$$
$$\left. -w_7\frac{(-1)^{n+\text{ind}g}}{n-2}\Theta_{[a\mathbf{n}][\mu\gamma]}{}^{\gamma}{}_{\nu}\left(*R^{\nu\mu}\right)\right],$$
(A.19)

where we used the variational results (A.11) and (A.12). We already used the equations of motion for $\varphi^{\mu\nu}$ that are $\varrho_{\mu\nu} = R_{\mu\nu}$. As next step, we need to $3+1$ decompose the curvature

indices by means of (41a) and impose the Gauß-Codazzi equations to replace the curvature decomposition terms. We note that $R_{\mu\nu} = W_{\mu\nu}$ is implicit in the irreducible components $^{(I)}W_{\mu\nu}$ of $W_{\mu\nu}$. For the considered case of vanishing non-metricity, the generalized Gauß-Codazzi equations (11) simplify to

$$
\begin{aligned}
e_\mu^a e_b^\nu R^\mu{}_\nu &= -\mathrm{d}\omega_b^a - \omega_c^a \wedge \omega_b^c + \varepsilon K^a \wedge K_b\,, \\
n_\mu e_a^\nu R^\mu{}_\nu &= -e_a^\mu n^\nu R_{\mu\nu} = DK^a\,,
\end{aligned}
\tag{A.20}
$$

where the extrinsic curvature one-form $K^a$ was defined in (4).

Using (12) we finally write the GHY term of (A.18) as

$$
S_{\mathrm{GHY\ MAG},\rho}^{Q=0} = 2\varepsilon \int_{\partial\mathcal{M}} K^a \wedge *\varphi_{\mathbf{n}a}\big|_{\varrho=R,\,\text{Gauß-Codazzi}}\big|_{\partial\mathcal{M}}\,,
\tag{A.21}
$$

where $*\varphi_{\mathbf{n}a}\big|_{\varrho=R,\,\text{Gauß-Codazzi}}$ is understood as evaluation of (A.19) using the Gauß-Codazzi equations (A.20). This evaluation is straightforward but yet obviously cumbersome so we state only the implicit result here and leave the full evaluation to computer algebra systems. The same holds for the generalization to the full MAG Lagrangian in presence of non-metricity.

## B   GHY term for metric compatible theories

In section 2 we comment on the calculation of the GHY term in the case of vanishing non-metricity, $Q_{\mu\nu} \equiv -Dg_{\mu\nu} = 0$. The latter condition enforces the curvature two-form $\Omega^\mu{}_\nu$ to be antisymmetric which is obtained from the Bianchi identity (2) of non-metricity, $\Omega_{\mu\nu} + \Omega_{\nu\mu} = DQ_{\mu\nu}$. The Lagrange multipliers $\varphi_{\mu\nu}$ and $\varrho_{\mu\nu}$ are assumed to have the same symmetries as $\Omega_{\mu\nu}$, but we recommend to consider $\varrho_{\mathbf{n}a}$ and $\varrho_{a\mathbf{n}}$ as being independent for the variational calculation in section 2 nevertheless. In the current section we examine the differences if one does not follow this method for the calculation but uses $\varrho_{\mathbf{n}a} = -\varrho_{a\mathbf{n}}$ in the variational process instead.

As a first step, we need to employ the antisymmetry of $\varrho_{\mu\nu}$ in the $3+1$ decomposition of $*\varphi^{\mu\nu} \wedge \delta\varrho_{\mu\nu}$. In contrast to the constraints for the calculation of $\varphi_{\mu\nu}$ we have found in (10) we now obtain

$$
*\varphi^{\mathbf{n}a} \wedge \delta\varrho_{\mathbf{n}a} = \frac{\varepsilon}{2}\delta_{\varrho_{\mathbf{n}a}}\mathcal{L}(\varrho_{\mathbf{n}a},\dots)
\tag{B.1}
$$

as the only relevant constraint. Note the factor $\frac{1}{2}$ on the right hand side of (B.1) in contrast to (10). This factor results from using the $\varrho_{\mu\nu}$ antisymmetry. By the same argument we obtain a factor of 2 from using the antisymmetry of $\varrho_{\mu\nu}$ in the calculation of $\delta_{\varrho_{\mathbf{n}a}}\mathcal{L}(\varrho_{\mathbf{n}a},\dots)$ for a specific theory. This factor of 2 cancels the factor $\frac{1}{2}$ in (B.1). For this reason we recommended to do the variational calculus without assuming the antisymmetry of $\varrho_{\mu\nu}$ in section 2. Note that this assumption is made only in the variational calculus when calculating $\varphi^{\mathbf{n}a}$ by means of the constraints (10). It can be considered as a method which simplifies the calculation.

Of course, for the asymmetry method to be valid, the GHY terms in both methods need to coincide. Indeed, using the antisymmetry of $\Omega^\mu{}_\nu$ and $\varphi_{\mu\nu}$ yields

$$
S_{\mathrm{GHY}}^{Q=0} = 2 \int_{\partial\mathcal{M}} \varepsilon K^a \wedge *\varphi_{\mathbf{n}a}\big|_{\partial\mathcal{M}}
\tag{B.2}
$$

as GHY term in coincidence with the result (13) from the asymmetry assumption. The discussion in this appendix generalizes straightforwardly if non-metricity is not vanishing.



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
