# Peer review of "Universal Gibbons-Hawking-York term for theories with curvature, torsion and non-metricity"

_SciPost Physics, doi:SciPost Phys. 14, 099 (2023)_

## Round 2 · Referee Report · Anonymous (Referee 1) · 2023-1-12

Strengths

1. Presents a general methods for deriving GHY terms.

2. Exemplifies/verifies the result in Einstein-Hilbert gravity, Chern-Simons modified gravity and Lovelock-Chern-Simons gravity.

3, The disposition of the presentation is very clear and pedagogical.

Weaknesses

No significant weaknesses

Report

The acceptance criteria are met.

Requested changes

No changes required

Attachment

---

## Round 2 · Referee Report · Anonymous (Referee 2) · 2023-1-18

Strengths

1. Has a universal, model-independent scope

2. Solves a very specific technical problem, namely how to derive Gibbons-Hawking-York-like boundary terms, for generic gravity(-like) theories

3. Uses minimal input and provides maximal generality, including applications to more exotic, higher-derivative theories and/or theories with torsion/non-metricity

Weaknesses

1. Provides no genuinely new example not considered in the previous literature, except for results relegated to an appendix (and the addition of torsion in the Lovelock example)

2. Not sure this is a weakness per se, but the two examples provided both differ by some numerical factor from corresponding earlier results in the literature

3. Parts of the paper are repetitive since there is a fairly long summary in section 2 (4 pages), and the actual derivation comes only after the two examples for this derivation

Report

The universality of the results presented in this paper could make this a standard reference for future purposes, especially for researchers interested in holography applied to metric affine gravity theories. While the Gibbons-Hawking-York boundary term is only one ingredient for this purpose, it is undoubtedly an important ingredient. It could have been nice to also include a generalization to supergravity, but the paper is worthwhile as it is.

In my opinion, the paper is suitable for publication in SciPost Physics, both in terms of topic and quality.

Requested changes

The paper could be published in SciPost Physics in its present form. I would have preferred a more condensed version of the presentation, basically eliminating section 2 and inserting instead section 4, and then ending with the examples of section 3. However, I realize that this is a matter of taste, so this should not be considered a requested change but merely an optional suggestion.

---

## Round 2 · Referee Report · Anonymous (Referee 3) · 2023-1-25

Strengths

1) Potentially interesting direction both in theoretical as well in applied (i.e. condensed matter) research research.

2) Very clear presentation of the technical points.

Weaknesses

1) One would have expected at least one simple application of their result i.e. a boundary one-point function of an exact bulk solution with torsion and nonmetricity.

Report

This work is relevant, well written and can be published in its present form. I urge the authors to consider continuing their research in this direction and think of specific examples.

Requested changes

No changes required.

---

## Editorial Decision

published